# TOWARDS FOUNDATION MODELS FOR KNOWLEDGE GRAPH REASONING

**Mikhail Galkin[1]**[*]**Xinyu Yuan[2,3], Hesham Mostafa[1], Jian Tang[2,4], Zhaocheng Zhu[2,3]**
[1]Intel AI Lab, [2]Mila [3]University of Montréal [4]HEC Montréal & CIFAR AI Chair

## ABSTRACT

Foundation models in language and vision have the ability to run inference on any textual and visual inputs thanks to the transferable representations such as a vocabulary of tokens in language. Knowledge graphs (KGs) have different entity and relation vocabularies that generally do not overlap. The key challenge of designing foundation models on KGs is to learn such transferable representations that enable inference on any graph with arbitrary entity and relation vocabularies. In this work, we make a step towards such foundation models and present ULTRA, an approach for learning universal and transferable graph representations. ULTRA builds relational representations as a function conditioned on their interactions. Such a conditioning strategy allows a pre-trained ULTRA model to inductively generalize to any unseen KG with any relation vocabulary and to be fine-tuned on any graph. Conducting link prediction experiments on 57 different KGs, we find that the *zero-shot* inductive inference performance of a single pre-trained ULTRA model on unseen graphs of various sizes is often on par or better than strong baselines trained on specific graphs. Fine-tuning further boosts the performance. The code is available: https://github.com/DeepGraphLearning/ULTRA.

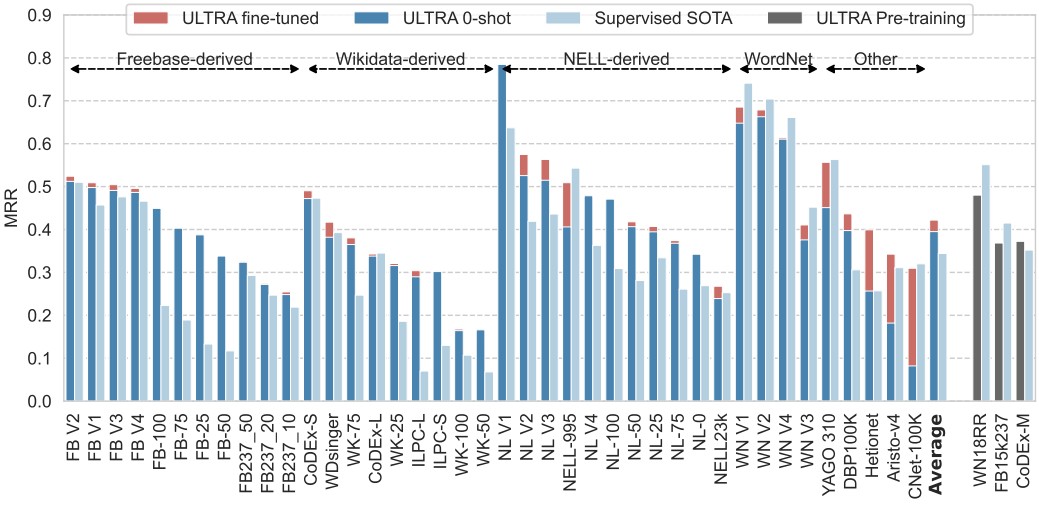

Figure 1: Zero-shot and fine-tuned MRR (higher is better) of ULTRA pre-trained on three graphs (FB15k-237, WN18RR, CoDEx-Medium). On average, zero-shot performance is better than best reported baselines trained on specific graphs (0.395 vs 0.344). More results in Figure 4 and Table 1.

## 1 INTRODUCTION

Modern machine learning applications increasingly rely on the *pre-training* and *fine-tuning* paradigm. In this paradigm, a backbone model often trained on large datasets in a self-supervised fashion is commonly known as a *foundation model* (FM) (Bommasani et al., 2021). After pre-training, FMs can be fine-tuned on smaller downstream tasks. In order to transfer to a broad set of downstream tasks, FMs leverage certain *invariances* pertaining to a domain of interest, *e.g.*, large

---

[*]Correspondence: mikhail.galkin@intel.com

language models like BERT (Devlin et al., 2019), GPT-4 (OpenAI, 2023), Llama-2 (Touvron et al., 2023) operate on a fixed vocabulary of tokens; vision models operate on raw pixels (He et al., 2016; Radford et al., 2021) or image patches (Dosovitskiy et al., 2021); chemistry models (Ying et al., 2021; Zheng et al., 2023) learn a vocabulary of atoms from the periodic table.

Representation learning on knowledge graphs (KGs), however, has not yet witnessed the benefits of transfer learning despite a wide range of downstream applications such as precision medicine (Chandak et al., 2023), materials science (Venugopal et al., 2022; Statt et al., 2023), virtual assistants (Ilyas et al., 2022), or product graphs in e-commerce (Dong, 2018). The key problem is that different KGs typically have different entity and relation vocabularies. Classic *transductive* KG embedding models (Ali et al., 2021) learn entity and relation embeddings tailored for each specific vocabulary and cannot generalize even to new nodes within the same graph. More recent efforts towards generalization across the vocabularies are known as *inductive* learning methods (Chen et al., 2023). Most of the inductive methods (Teru et al., 2020; Zhu et al., 2021; Galkin et al., 2022b; Zhang & Yao, 2022) generalize to new entities at inference time but require a fixed relation vocabulary to learn entity representations as a function of the relations. Such inductive methods still cannot transfer to KGs with a different set of relations, *e.g.*, training on Freebase and inference on Wikidata.

The main research goal of this work is *finding the invariances transferable across graphs with arbitrary entity and relation vocabularies*. Leveraging and learning such invariances would enable the *pre-train and fine-tune* paradigm of foundation models for KG reasoning where a single model trained on one graph (or several graphs) with one set of relations would be able to *zero-shot* transfer to any new, unseen graph with a completely different set of relations and relational patterns. Our approach to the problem is based on two key observations: (1) even if relations vary across the datasets, the interactions between the relations may be similar and transferable; (2) initial relation representations may be conditioned on this interaction bypassing the need for any input features. To this end, we propose ULTRA, a method for unified, learnable, and transferable KG representations that leverages the invariance of the *relational structure* and employs relative relation representations on top of this structure for parameterizing any unseen relation. Given any multi-relational graph, ULTRA first constructs a graph of relations (where each node is a relation from the original graph) capturing their interactions. Applying a graph neural network (GNN) with a *labeling trick* (Zhang et al., 2021) over the graph of relations, ULTRA obtains a unique *relative* representation of each relation. The relation representations can then be used by any inductive learning method for downstream applications like KG completion. Since the method does not learn any graph-specific entity or relation embeddings nor requires any input entity or relation features, ULTRA enables *zero-shot* generalization to any other KG of any size and any relational vocabulary.

Experimentally, we show that ULTRA paired with the NBFNet (Zhu et al., 2021) link predictor pretrained on three KGs (FB15k-237, WN18RR, and CoDEx-M derived from Freebase, WordNet, and Wikidata, respectively) generalizes to 50+ different KGs with sizes of 1,000–120,000 nodes and 5K–1M edges. ULTRA demonstrates promising transfer learning capabilities where the zero-shot inference performance on those unseen graphs might exceed strong supervised baselines by up to 300%. The subsequent short fine-tuning of ULTRA often boosts the performance even more.

## 2 RELATED WORK

**Inductive Link Prediction.** In contrast to transductive methods that only support a fixed set of entities and relations during training, inductive methods (Chen et al., 2023) aim at generalizing to graphs with unseen nodes (with the same set of relations) or to both new entities and relations. The majority of existing inductive methods such as GraIL (Teru et al., 2020), NBFNet (Zhu et al., 2021), NodePiece (Galkin et al., 2022b), RED-GNN (Zhang & Yao, 2022) can generalize to graphs only with new nodes, but not to new relation types since the node representations are constructed as a function of the fixed relational vocabulary.

First approaches that support unseen relations at inference resorted to meta-learning and few-shot learning (Chen et al., 2019; Zhang et al., 2020; Huang et al., 2022). Meta-learning is computationally expensive and is hardly scalable to large graphs. Few-shot learning methods do not work on the whole new unseen inference graph but instead mine many *support sets* akin to subgraph sampling.

Both RMPI (Geng et al., 2023) and InGram (Lee et al., 2023) employ graphs of relations to generalize to unseen domains. However, RMPI suffers from the same computational and scalability issues

as subgraph sampling methods. InGram is more scalable but its featurization strategy relies on the discretization of node degrees that only transfers to graphs of a similar relational distribution and does not transfer to arbitrary KGs. Gao et al. (2023) introduce the notion of *double equivariance*, *i.e.*, relation exchangeability in multi-relational graphs, as a general theoretical framework for inductive reasoning that transfers to any relations at inference. ISDEA (Gao et al., 2023) is the first approach to design doubly equivariant GNNs and MTDEA (Zhou et al., 2023) further extends the theory to partial equivariance. However, ISDEA and MTDEA are computationally expensive and cannot scale to graphs considered in this work. Similarly to RMPI, InGram, ISDEA, and MTDEA, ULTRA transfers to *any* unseen KG in the zero-shot fashion, but exhibits better generalization capabilities, scales to graphs of millions of edges, and introduces only a marginal inference overhead (one-step pre-computation) to any inductive link predictor.

**Text-based methods.** A line of inductive link prediction methods like BLP (Daza et al., 2021), KEPLER (Wang et al., 2021), StATIK (Markowitz et al., 2022), RAILD (Gesese et al., 2022) rely on textual descriptions of entities and relations and use language models to encode them. PRODIGY (Huang et al., 2023a) uses text features for few-shot node classification tasks. We deem this family of methods orthogonal to ULTRA as we assume the graphs do not have any input features and leverage only structural information encoded in the graph. Furthermore, the zero-shot inductive transfer to an arbitrary KG studied in this work implies running inference on graphs from different domains that might need different language encoders, *e.g.*, models trained on general English data are unlikely to transfer to graphs with descriptions in other languages or domain-specific graphs.

## 3 PRELIMINARIES

**Knowledge Graph and Inductive Learning.** Given a finite set of entities $\mathcal{V}$ (nodes), a finite set of relations $\mathcal{R}$ (edge types), and a set of triples (edges) $\mathcal{E} = (\mathcal{V} \times \mathcal{R} \times \mathcal{V})$, a knowledge graph $\mathcal{G}$ is a tuple $\mathcal{G} = (\mathcal{V}, \mathcal{R}, \mathcal{E})$. In the *transductive* setup, the graph at training time $\mathcal{G}_{train} = (\mathcal{V}_{train}, \mathcal{R}_{train}, \mathcal{E}_{train})$ and the graph at inference (validation or test) time $\mathcal{G}_{inf} = (\mathcal{V}_{inf}, \mathcal{R}_{inf}, \mathcal{E}_{inf})$ are the same, *i.e.*, $\mathcal{G}_{train} = \mathcal{G}_{inf}$. In the *inductive* setup, in the general case, the training and inference graphs are different, $\mathcal{G}_{train} \neq \mathcal{G}_{inf}$. In the easier setup tackled by most of the literature, the relation set $\mathcal{R}$ is fixed and shared between training and inference graphs, *i.e.*, $\mathcal{G}_{train} = (\mathcal{V}_{train}, \mathcal{R}, \mathcal{E}_{train})$ and $\mathcal{G}_{inf} = (\mathcal{V}_{inf}, \mathcal{R}, \mathcal{E}_{inf})$. The inference graph can be an extension of the training graph if $\mathcal{V}_{train} \subseteq \mathcal{V}_{inf}$ or be a separate disjoint graph (with the same set of relations) if $\mathcal{V}_{train} \cap \mathcal{V}_{inf} = \emptyset$. In the hardest inductive case, both entities and relations sets are different, *i.e.*, $\mathcal{V}_{train} \cap \mathcal{V}_{inf} = \emptyset$ and $\mathcal{R}_{train} \cap \mathcal{R}_{inf} = \emptyset$. In this work, we tackle this harder inductive (also known as *fully-inductive*) case with both new, unseen entities and relation types at inference time. Since the harder inductive case (with new relations at inference) is strictly a superset of the easier inductive scenario (with the fixed relation set), any model capable of fully-inductive inference is by design applicable in easier inductive scenarios as well.

**Problem Formulation.** Each triple $(h, r, t) \in (\mathcal{V} \times \mathcal{R} \times \mathcal{V})$ denotes a head entity $h$ connected to a tail entity $t$ by relation $r$. The knowledge graph reasoning task answers queries $(h, r, ?)$ or $(?, r, t)$. It is common to rewrite the head-query $(?, r, t)$ as $(t, r^{-1}, ?)$ where $r^{-1}$ is the inverse relation of $r$. The set of target triples $\mathcal{E}_{pred}$ is predicted based on the incomplete inference graph $\mathcal{G}_{inf}$ which is a part of the unobservable complete graph $\hat{\mathcal{G}}_{inf} = (\mathcal{V}_{inf}, \mathcal{R}_{inf}, \hat{\mathcal{E}}_{inf})$ where $\hat{\mathcal{E}}_{inf} = \mathcal{E}_{inf} \cup \mathcal{E}_{pred}$.

**Link Prediction and Labeling Trick GNNs.** Standard GNN encoders (Kipf & Welling, 2017; Veličković et al., 2018) including those for multi-relational graphs (Vashishth et al., 2020) underperform in link prediction tasks due to neighborhood symmetries (*automorphisms*) that assign different (but *automorphic*) nodes the same features making them indistinguishable. To break those symmetries, *labeling tricks* (Zhang et al., 2021) were introduced that assign each node a unique feature vector based on its structural properties. Most link predictors that use the labeling tricks (Teru et al., 2020; Zhang et al., 2021; Chamberlain et al., 2023) mine numerical features like Double Radius Node Labeling (Zhang & Chen, 2018) or Distance Encoding (Li et al., 2020). In contrast, multi-relational models like NBFNet (Zhu et al., 2021) leverage an indicator function $\text{INDICATOR}(h, v, r)$ and label (initialize) the head node $h$ with the query vector $\boldsymbol{r}$ that can be learned while other nodes $v$ are initialized with zeros. In other words, final node representations are *conditioned* on the query relation and NBFNet learns *conditional node representations*. Conditional representations were shown to be provably more expressive theoretically (Huang et al., 2023b) and practically effective (Zhu et al., 2022a; Galkin et al., 2022c) than standard unconditional GNN encoders.

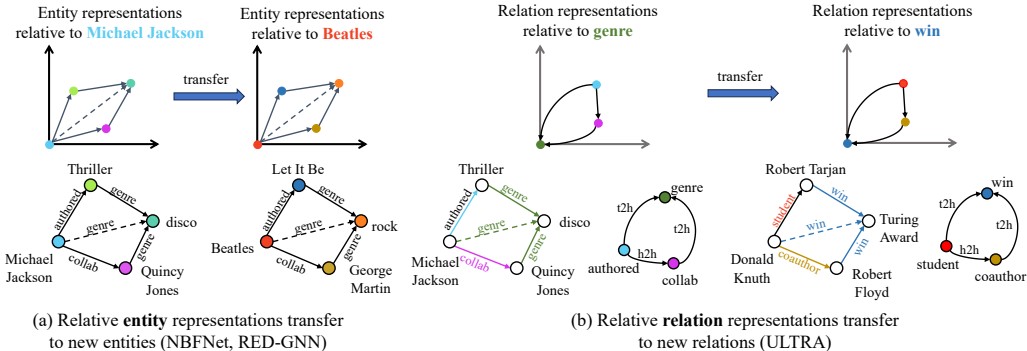

Figure 2: (a) relative **entity** representations used in inductive models generalize to new entities; (b) relative **relation** representations based on a graph of relations generalize to both new relations and entities. The graph of relations captures four fundamental interactions (*t2h*, *h2h*, *h2t*, *h2h*) independent from any graph-specific relation vocabulary and whose representations can be learned.

## 4 METHOD

The key challenge of inductive inference with different entity and relation vocabularies is finding *transferable invariances* that would produce entity and relation representations conditioned on the new graph (as learning entity and relation embedding matrices from the training graph is useless and not transferable). Most inductive GNN methods that transfer to new entities (Zhu et al., 2021; Zhang & Yao, 2022) learn **relative entity representations** conditioned on the graph structure as shown in Fig. 2 (a). For example, given $a, b, c$ are variable entities and $a$ as a root node labeled with INDICATOR(), a structure $a \xrightarrow{authored} b \xrightarrow{genre} c \wedge a \xrightarrow{collab} d \xrightarrow{genre} c$ might imply existence of the edge $a \xrightarrow{genre} c$. Learning such a structure on a training set with entities *Michael Jackson* $\xrightarrow{authored}$ *Thriller* $\xrightarrow{genre}$ *disco* seamlessly transfers to new entities *Beatles* $\xrightarrow{authored}$ *Let It Be* $\xrightarrow{genre}$ *rock* at inference time without learning entity embeddings thanks to the same relational structure and *relative* entity representations. As training and inference relations are the same $\mathcal{R}_{train} = \mathcal{R}_{inf}$, such approaches learn relation embedding matrices and use **relations as invariants**.

In ULTRA, we generalize KG reasoning to both new entities and relations (where $\mathcal{R}_{train} \neq \mathcal{R}_{inf}$) by leveraging a *graph of relations*, *i.e.*, a graph where each node corresponds to a distinct relation type[1] in the original graph. While relations at inference time are different, their interactions remain the same and are captured by the graph of relations. For example, Fig. 2 (b), a *tail* node of the *authored* relation is also a *head* node of the *genre* relation. Hence, *authored* and *genre* nodes are connected by the *tail-to-head* edge in the relation graph. Similarly, *authored* and *collab* share the same *head* node in the entity graph and thus are connected with the *head-to-head* edge in the relation graph. Overall, we distinguish **four** such core, *fundamental* relation-to-relation interactions[2]: *tail-to-head (t2h)*, *head-to-head (h2h)*, *head-to-tail (h2t)*, and *tail-to-tail (t2t)*. Albeit relations in the inference graph in Fig. 2 (b) are different, their graph of relations and relation interactions resemble that of the training graph. Hence, we could leverage the **invariance of the relational structure** and four fundamental relations to obtain relational representations of the unseen inference graph. As a typical KG reasoning task $(h, q, ?)$ is conditioned on a query relation $q$, it is possible to build representations of all relations *relative* to the query $q$ by using a labeling trick on top of the graph of relations. Such **relative relation representations** do not need any input features and naturally generalize to any multi-relational graph.

Practically (Fig. 3), given a query $(h, q, ?)$ over a graph $\mathcal{G}$, ULTRA employs a three-step algorithm that we describe in the following subsections. (1) Lift the original graph $\mathcal{G}$ to the graph of relations $\mathcal{G}_r$ – Section 4.1; (2) Obtain relative relation representations $\mathbf{R}_q | (q, \mathcal{G}_r)$ conditioned on the query relation $q$ in the relation graph $\mathcal{G}_r$ – Section 4.2; (3) Using the relation representations $\mathbf{R}_q$ as starting relation features, run inductive link prediction on the original graph $\mathcal{G}$ – Section 4.3.

---

[1] We also add inverse relations as nodes to the relation graph.

[2] Other strategies for capturing relation-to-relation interactions might exist beside those four types and we leave their exploration for future work.

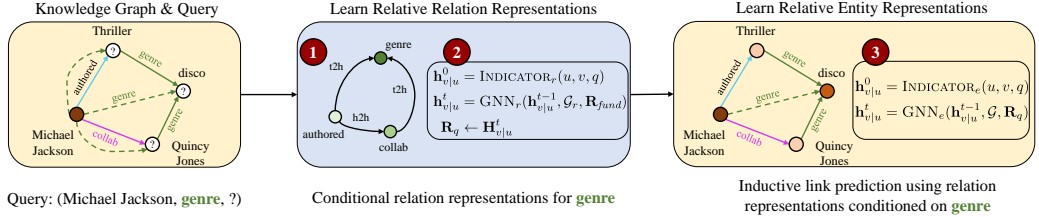

Figure 3: Given a query $(h, q, ?)$ on graph $\mathcal{G}$, ULTRA (1) builds a graph of relations $\mathcal{G}_r$ with four interactions $\mathcal{R}_{fund}$ (Sec. 4.1); (2) builds relation representations $\boldsymbol{R}_q$ conditioned on the query relation $q$ and $\mathcal{G}_r$ (Sec. 4.2); (3) runs any inductive link predictor on $\mathcal{G}$ using representations $\boldsymbol{R}_q$ (Sec. 4.3).

## 4.1 RELATION GRAPH CONSTRUCTION

Given a graph $\mathcal{G} = (\mathcal{V}, \mathcal{R}, \mathcal{E})$, we first apply the lifting function $\mathcal{G}_r = \text{LIFT}(\mathcal{G})$ to build a graph of relations $\mathcal{G}_r = (\mathcal{R}, \mathcal{R}_{fund}, \mathcal{E}_r)$ where each node is a distinct relation type[3] in $\mathcal{G}$. Edges $\mathcal{E}_r \in (\mathcal{R} \times \mathcal{R}_{fund} \times \mathcal{R})$ in the relation graph $\mathcal{G}_r$ denote interactions between relations in the original graph $\mathcal{G}$, and we distinguish four such fundamental relation interactions $\mathcal{R}_{fund}$: *tail-to-head (t2h)* edges, *head-to-head (h2h)* edges, *head-to-tail (h2t)* edges, and *tail-to-tail (t2t)* edges. The full adjacency tensor of the relation graph is $\boldsymbol{A}_r \in \mathbb{R}^{|\mathcal{R}| \times |\mathcal{R}| \times 4}$. Each of the four adjacency matrices can be efficiently obtained with one sparse matrix multiplication (Appendix B).

## 4.2 CONDITIONAL RELATION REPRESENTATIONS

Given a query $(h, q, ?)$ and a relation graph $\mathcal{G}_r$, we then obtain $d$-dimensional node representations $\boldsymbol{R}_q \in \mathbb{R}^{|\mathcal{R}| \times d}$ of $\mathcal{G}_r$ (corresponding to all edge types $\mathcal{R}$ in the original graph $\mathcal{G}$) conditioned on the query relation $q$. Practically, we implement conditioning by applying a labeling trick to initialize the node $q$ in $\mathcal{G}_r$ through the INDICATOR$_r$ function and employ a message passing GNN over $\mathcal{G}_r$:

$$\boldsymbol{h}_{v|q}^0 = \text{INDICATOR}_r(v, q) = \mathbb{1}_{v=q} * \boldsymbol{1}^d, \quad v \in \mathcal{G}_r$$

$$\boldsymbol{h}_{v|q}^{t+1} = \text{UPDATE}\Big(\boldsymbol{h}_{v|q}^t, \text{AGGREGATE}\big(\text{MESSAGE}(\boldsymbol{h}_{w|q}^t, \boldsymbol{r})|w \in \mathcal{N}_r(v), r \in \mathcal{R}_{fund}\big)\Big)$$

The indicator function is implemented as INDICATOR$_r(v, q) = \mathbb{1}_{v=q} * \boldsymbol{1}^d$ that simply puts a vector of ones on a node $v$ corresponding to the query relation $q$, and zeros otherwise. Following Huang et al. (2023b), we found that all-ones labeling with $\boldsymbol{1}^d$ generalizes better to unseen graphs of various sizes than a learnable vector. The GNN architecture (denoted as GNN$_r$ as it operates on the relation graph $\mathcal{G}_r$) follows NBFNet (Zhu et al., 2021) with a non-parametric DistMult (Yang et al., 2015) message function and sum aggregation. The only learnable parameters in each layer are embeddings of four fundamental interactions $\boldsymbol{R}_{fund} \in \mathbb{R}^{4 \times d}$, a linear layer for the UPDATE function, and an optional layer normalizaiton. Note that our general setup (Section 3) assumes no given input entity or relation features, so our parameterization strategy can be used to obtain relational representations of *any* multi-relational graph.

To sum up, each unique relation $q \in \mathcal{R}$ in the query has its own matrix of conditional relation representations $\boldsymbol{R}_q \in \mathbb{R}^{|\mathcal{R}| \times d}$ used by the entity-level reasoner for downstream applications.

## 4.3 ENTITY-LEVEL LINK PREDICTION

Given a query $(h, q, ?)$ over a graph $\mathcal{G}$ and conditional relation representations $\boldsymbol{R}_q$ from the previous step, it is now possible to adapt any off-the-shelf inductive link predictor that only needs relational features (Zhu et al., 2021; Zhang & Yao, 2022; Zhu et al., 2023; Zhang et al., 2023) to balance between performance and scalability. We modify another instance of NBFNet (GNN$_e$ as it operates on the entity level) to account for separate relation representations per query:

$$\boldsymbol{h}_{v|u}^0 = \text{INDICATOR}_e(u, v, q) = \mathbb{1}_{u=v} * \boldsymbol{R}_q[q], \quad v \in \mathcal{G}$$

$$\boldsymbol{h}_{v|u}^{t+1} = \text{UPDATE}\Big(\boldsymbol{h}_{v|u}^t, \text{AGGREGATE}\big(\text{MESSAGE}(\boldsymbol{h}_{w|u}^t, g^{t+1}(\boldsymbol{r}))|w \in \mathcal{N}_r(v), r \in \mathcal{R}\big)\Big)$$

---

[3] $2|\mathcal{R}|$ nodes after adding inverse relations to the original graph.

That is, we first initialize the head node $h$ with the query vector $q$ from $\boldsymbol{R}_q$ whereas other nodes are initialized with zeros. Each $t$-th GNN layer applies a non-linear function $g^t(\cdot)$ to transform original relation representations to layer-specific relation representations as $\boldsymbol{R}^t = g^t(\boldsymbol{R}_q)$ from which the edge features are taken for the MESSAGE function. $g(\cdot)$ is implemented as a 2-layer MLP with ReLU. Similar to $\text{GNN}_r$ in Section 4.2, we use sum aggregation and a linear layer for the UPDATE function. After message passing, the final MLP $s : \mathbb{R}^d \rightarrow \mathbb{R}^1$ maps the node states to logits $p(h, q, v)$ denoting the score of a node $v$ to be a tail of the initial query $(h, q, ?)$.

**Training.**  ULTRA can be trained on any multi-relational graph or mixture of graphs thanks to the inductive and conditional relational representations. Following the standard practices in the literature (Sun et al., 2019; Zhu et al., 2021), ULTRA is trained by minimizing the binary cross entropy loss over positive and negative triplets

$$\mathcal{L} = -\log p(u, q, v) - \sum_{i=1}^{n} \frac{1}{n} \log(1 - p(u_i', q, v_i'))$$

where $(u, q, v)$ is a positive triple in the graph and $\{(u_i', q, v_i')\}_{i=1}^{n}$ are negative samples obtained by corrupting either the head $u$ or tail $v$ of the positive sample.

## 5 EXPERIMENTS

To evaluate the qualities of ULTRA as a foundation model for KG reasoning, we explore the following questions: (1) Is pre-trained ULTRA able to inductively generalize to unseen KGs in the zero-shot manner? (2) Are there any benefits from fine-tuning ULTRA on a specific dataset? (3) How does a single pre-trained ULTRA model compare to models trained from scratch on each target dataset? (4) Do more graphs in the pre-training mix correspond to better performance?

### 5.1 SETUP AND DATASETS

**Datasets.**  We conduct a broad evaluation on 57 different KGs with reported, non-saturated results on the KG completion task. The datasets can be categorized into three groups:

- *Transductive* datasets (16 graphs) with the fixed set of entities and relations at training and inference time ($\mathcal{G}_{train} = \mathcal{G}_{inf}$): FB15k-237 (Toutanova & Chen, 2015), WN18RR (Dettmers et al., 2018), YAGO3-10 (Mahdisoltani et al., 2014), NELL-995 (Xiong et al., 2017), CoDEx (Small, Medium, and Large) (Safavi & Koutra, 2020), WDsinger, NELL23k, FB15k237(10), FB15k237(20), FB15k237(50) (Lv et al., 2020), AristoV4 (Chen et al., 2021), DBpedia100k (Ding et al., 2018), ConceptNet100k (Malaviya et al., 2020), Hetionet (Himmelstein et al., 2017)
- *Inductive entity* ($e$) datasets (18 graphs) with new entities at inference time but with the fixed set of relations ($\mathcal{V}_{train} \neq \mathcal{V}_{inf}, \mathcal{R}_{train} = \mathcal{R}_{inf}$): 12 datasets from GraIL (Teru et al., 2020), 4 graphs from INDIGO (Liu et al., 2021; Hamaguchi et al., 2017), and 2 ILPC 2022 datasets (Small and Large) (Galkin et al., 2022a).
- *Inductive entity and relation* ($e, r$) datasets (23 graphs) where both entities and relations at inference are new ($\mathcal{V}_{train} \neq \mathcal{V}_{inf}, \mathcal{R}_{train} \neq \mathcal{R}_{inf}$): 13 graphs from INGRAM (Lee et al., 2023) and 10 graphs from MTDEA (Zhou et al., 2023).

In practice, however, a pre-trained ULTRA operates in the *inductive* $(e, r)$ mode on all datasets (apart from those in the training mixture) as their sets of entities, relations, and relational structures are different from the training set. The dataset sizes vary from 1k to 120k entities and 1k-2M edges in the inference graph. We provide more details on the datasets in Appendix A.

**Pretraining and Fine-tuning.**  ULTRA is pre-trained on the mixture of 3 standard KGs (WN18RR, CoDEx-Medium, FB15k237) to capture the variety of possible relational structures and sparsities in respective relational graphs $\mathcal{G}_r$. ULTRA is relatively small (177k parameters in total, with 60k parameters in $\text{GNN}_r$ and 117k parameters in $\text{GNN}_e$) and is trained for 200,000 steps with batch size of 64 with AdamW optimizer on 2 A100 (40 GB) GPUs. All fine-tuning experiments were done on a single RTX 3090 GPU. More details on hyperparameters and training are in Appendix C.

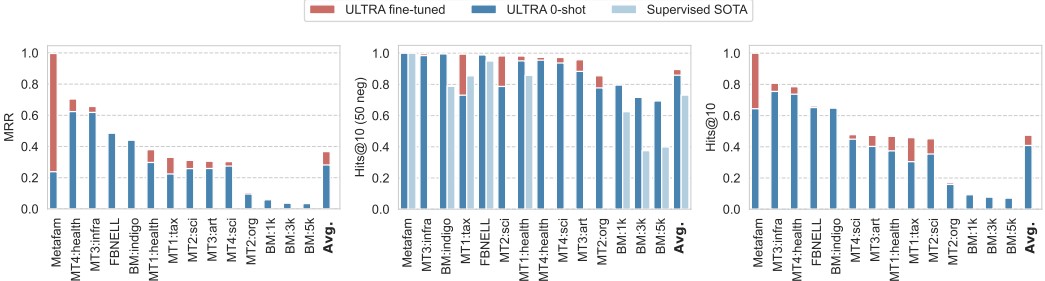

Figure 4: ULTRA performance on 14 inductive datasets from MTDEA (Zhou et al., 2023) and INDIGO (Liu et al., 2021) for 8 of which only an approximate metric *Hits@10 (50 negs)* is available (center). We also report full MRR (left) and Hits@10 (right) computed on the entire entity sets demonstrating that Hits@10 (50 negs) overestimates the real performance.

Table 1: Zero-shot and fine-tuned performance of ULTRA compared to the published supervised SOTA on 51 datasets (as in Fig. 1 and Fig. 4). The zero-shot ULTRA outperforms supervised baselines on average and on inductive datasets. Fine-tuning improves the performance even further. We report pre-training performance to the fine-tuned version. More detailed results are in Appendix D.

| Model | Inductive $(e) + (e, r)$ (27 graphs) | | Transductive $e$ (13 graphs) | | Total Avg (40 graphs) | | Pretraining (3 graphs) | | Inductive $(e) + (e, r)$ (8 graphs) |
|---|---|---|---|---|---|---|---|---|---|
| | **MRR** | **H@10** | **MRR** | **H@10** | **MRR** | **H@10** | **MRR** | **H@10** | **Hits@10 (50 negs)** |
| Supervised SOTA | 0.342 | 0.482 | 0.348 | 0.494 | 0.344 | 0.486 | **0.439** | **0.585** | 0.731 |
| ULTRA 0-shot | 0.435 | 0.603 | 0.312 | 0.458 | 0.395 | 0.556 | - | - | 0.859 |
| ULTRA fine-tuned | **0.443** | **0.615** | **0.379** | **0.543** | **0.422** | **0.592** | 0.407 | 0.568 | **0.896** |

**Evaluation Protocol.** We report Mean Reciprocal Rank (MRR) and Hits@10 (H@10) as the main performance metrics evaluated against the full entity set of the inference graph. For each triple, we report the results of predicting both heads and tails. Only in three datasets from Lv et al. (2020) we report tail-only metrics similar to the baselines. In the zero-shot inference scenario, we run a pre-trained model on the inference graph and test set of triples. In the fine-tuning case, we further train the model on the training split of each dataset retaining the checkpoint of the best validation set MRR. We run zero-shot inference experiments once as the results are deterministic, and report an average of 5 runs for each fine-tuning run on each dataset.

**Baselines.** On each graph, we compare ULTRA against the reported state-of-the-art model (we list SOTA for all 57 graphs in Appendix A). To date, all of the reported SOTA models are trained end-to-end specifically on each target dataset. Due to the computational complexity of baselines, the only existing results on 4 MTDEA datasets (Zhou et al., 2023) and 4 INDIGO datasets (Liu et al., 2021) report Hits@10 against 50 randomly chosen negatives. We compare ULTRA against those baselines using this *Hits@10 (50 negs)* metric as well as report the full performance on the whole entity sets.

## 5.2 MAIN RESULTS: ZERO-SHOT INFERENCE AND FINE-TUNING OF ULTRA

The main experiment reports how ULTRA pre-trained on 3 graphs inductively generalizes to 54 other graphs both in the zero-shot (0-shot) and fine-tuned cases. Fig. 1 compares ULTRA with supervised SOTA baselines on 43 graphs that report MRR on the full entity set. Fig. 4 presents the comparison on the rest 14 graphs including 8 graphs for which the baselines report *Hits@10 (50 negs)*. The aggregated results on 51 graphs with available baseline results are presented in Table 1 and the complete evaluation on 57 graphs grouped into three families according to Section 5.1 is in Table 2. Full per-dataset results with standard deviations can be found in Appendix D.

On average, ULTRA outperforms the baselines even in the 0-shot inference scenario both in MRR and Hits@10. The largest gains are achieved on smaller inductive graphs, *e.g.*, on FB-25 and FB-50 0-shot ULTRA yields almost $3\times$ better performance (291% and 289%, respectively). During pre-training, ULTRA does not reach the baseline performance (0.407 vs 0.439 average MRR) and we link that with the lower 0-shot inference results on larger transductive graphs. However, fine-

Table 2: Zero-shot and fine-tuned ULTRA results on the complete set of 57 graphs grouped by the dataset category. Fine-tuning especially helps on larger transductive datasets and boosts the total average MRR by 10%. Additionally, we report as *(train e2e)* the average performance of dataset-specific ULTRA models trained from scratch on each graph. More detailed results are in Appendix D.

| Model | Inductive $e, r$ (23 graphs) | | Inductive $e$ (18 graphs) | | Transductive (13 graphs) | | Total Avg (54 graphs) | | Pretraining (3 graphs) | |
|---|---|---|---|---|---|---|---|---|---|---|
| | **MRR** | **H@10** | **MRR** | **H@10** | **MRR** | **H@10** | **MRR** | **H@10** | **MRR** | **H@10** |
| ULTRA (train e2e) | 0.392 | 0.552 | 0.402 | 0.559 | 0.384 | 0.545 | 0.393 | 0.552 | 0.403 | 0.562 |
| ULTRA 0-shot | 0.345 | 0.513 | 0.431 | 0.566 | 0.312 | 0.458 | 0.366 | 0.518 | - | - |
| ULTRA fine-tuned | 0.397 | 0.556 | 0.442 | 0.582 | 0.379 | 0.543 | 0.408 | 0.562 | 0.407 | 0.568 |

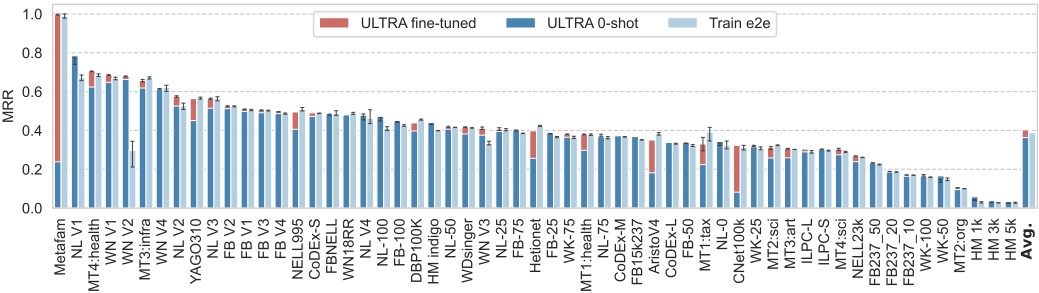

Figure 5: Comparison of zero-shot and fine-tuned ULTRA per-dataset performance against training a model from scratch on each dataset *(Train e2e)*. Zero-shot performance of a single pre-trained model is on par with training from scratch while fine-tuning yields overall best results.

tuning ULTRA effectively bridges this gap and surpasses the baselines. We hypothesize that in larger transductive graphs fine-tuning helps to adapt to different graph sizes (training graphs have 15-40k nodes while larger inference ones grow up to 123k nodes).

Following the sample efficiency and fast convergence of NBFNet (Zhu et al., 2021), we find that 1000-2000 steps are enough for fine-tuning ULTRA. In some cases (see Appendix D) fine-tuning brings marginal improvements or marginal negative effects. Averaged across 54 graphs (Table 2), fine-tuned ULTRA brings further 10% relative improvement over the zero-shot version.

## 5.3 ABLATION STUDY

We performed several experiments to better understand the pre-training quality of ULTRA and measure the impact of conditional relation representations on the performance.

**Positive transfer from pre-training.** We first study how a single pre-trained ULTRA model compares to training instances of the same model separately on each graph end-to-end. For that, for each of 57 graphs, we train 3 ULTRA instances of the same configuration and different random seeds until convergence and report the averaged results in Table 2 with per-dataset comparison in Fig. 5. We find that, on average, a single pre-trained ULTRA model in the zero-shot regime performs almost on par with the trained separate models, lags behind those on larger transductive graphs and exhibits better performance on inductive datasets. Fine-tuning a pre-trained ULTRA shows overall the best performance and requires significantly less computational resources than training a model from scratch on every target graph.

**Number of graphs in the pre-training mix.** We then study how inductive inference performance depends on the training mixture. While the main ULTRA model was trained on the mixture of three graphs, here we train more models varying the amount of KGs in the training set from a single FB15k237 to a combination of 8 transductive KGs (more details in Appendix C). For the fair comparison, we evaluate pre-trained models in the zero-shot regime only on inductive datasets (41 graphs overall). The results are presented in Fig. 6 where we observe the saturation of performance having more than three graphs in the mixture. We hypothesize that getting higher inference perfor-

Table 3: Ablation study: pre-training and zero-shot inference results of the main ULTRA, ULTRA without edge types in the relation graph (no etypes), ULTRA without edge types and with InGram-like (Lee et al., 2023) unconditional GNN over relation graph where nodes are initialized with all ones (ones) or with Glorot initialization (random). Averaged results over 3 categories of datasets.

| Model | Inductive $e, r$ (23 graphs) | | Inductive $e$ (18 graphs) | | Transductive (13 graphs) | | Total Avg (54 graphs) | | Pretraining (3 graphs) | |
|---|---|---|---|---|---|---|---|---|---|---|
| | MRR | H@10 | MRR | H@10 | MRR | H@10 | MRR | H@10 | MRR | H@10 |
| ULTRA | 0.345 | 0.513 | 0.431 | 0.566 | 0.312 | 0.458 | 0.366 | 0.518 | 0.407 | 0.568 |
| - no etypes in rel. graph | 0.292 | 0.466 | 0.389 | 0.539 | 0.258 | 0.409 | 0.316 | 0.477 | 0.357 | 0.517 |
| - no etypes, - uncond. GNN (ones) | 0.187 | 0.328 | 0.262 | 0.430 | 0.135 | 0.257 | 0.199 | 0.345 | 0.263 | 0.424 |
| - no etypes, - uncond. GNN (random) | 0.177 | 0.309 | 0.250 | 0.417 | 0.138 | 0.255 | 0.192 | 0.332 | 0.266 | 0.433 |

mance is tied up with model capacity, scale, and optimization. We leave that study along with more principled approached to selecting a pre-training mix for future work.

**Conditional vs unconditional relation graph encoding.** To measure the impact of the graph of relations and conditional relation representations, we pre-train three more models on the same mixture of three graphs varying several components: (1) we exclude four fundamental relation interactions (*h2h, h2t, t2h, t2t*) from the relation graph making it homogeneous and single-relational; (2) a homogeneous relation graph with an *unconditional* GNN encoder following the R-GATv2 architecture from the previous SOTA approach, InGram (Lee et al., 2023). The unconditional GNN needs input node features and we probed two strategies: Glorot initialization used in Lee et al. (2023) and initializing all nodes with a vector of ones $\mathbf{1}^d$.

The results are presented in Table 3 and indicate that ablated models struggle to reach the same pre-training performance and exhibit poor zero-shot generalization performance across all groups of graphs, *e.g.*, up to 48% relative MRR drop (0.192 vs 0.366) on the model with a homogeneous relation graph and randomly initialized node states with the unconditional R-GATv2 encoder. We therefore posit that conditional representations (both on relation and entity levels) are crucial for transferable representations for link prediction tasks that often require pairwise representations to break neighborhood symmetries.

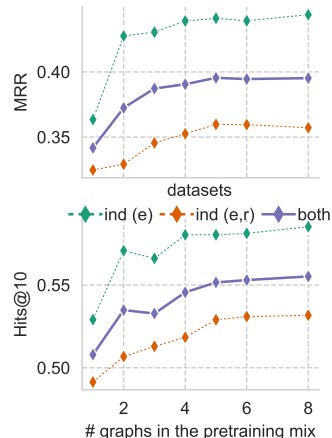

Figure 6: Averaged 0-shot performance on inductive datasets and # graphs in pre-training.

## 6 DISCUSSION AND FUTURE WORK

**Limitations and Future Work.** Albeit ULTRA demonstrates promising capabilities as a foundation model for KG reasoning in the zero-shot and fine-tuning regimes, there are several limitations and open questions. First, pre-training on more graphs does not often correspond to better inference performance. We hypothesize the reason might be in the overall small model size (177k parameters) and limited model capacity, *i.e.*, with increasing the diversity of training data the model size should increase as well. On the other hand, our preliminary experiments did not show significant improvements of scaling the parameter count beyond 200k. We hypothesize it might be an issue of input normalization and model optimization. We plan to address those open questions in the future work.

**Conclusion.** We presented ULTRA, an approach to learn universal and transferable graph representations that can serve as one of the methods towards building foundation models for KG reasoning. ULTRA enables training and inference on *any* multi-relational graph without any input features leveraging the invariance of the relational structure and conditional relation representations. Experimentally, a single pre-trained ULTRA model outperforms state-of-the-art tailored supervised baselines on 50+ graphs of 1k–120k nodes even in the *zero-shot* regime by average 15%. Fine-tuning ULTRA is sample-efficient and improves the average performance by further 10%. We hope that ULTRA contributes to the search for inductive and transferable representations where a single pre-trained model can inductively generalize to any graph and perform a variety of downstream tasks.

ETHICS STATEMENT

Foundation models can be run on tasks and datasets that were originally not envisioned by authors. Due to the ubiquitous nature of graph data, foundation graph models might be used for malicious activities like searching for patterns in anonymized data. On the other, more positive side, foundation models reduce the computational burden and carbon footprint of training many non-transferable graph-specific models. Having a single model with zero-shot transfer capabilities to any graph renders tailored graph-specific models unnecessary, and fine-tuning costs are still lower than training any model from scratch.

REPRODUCIBILITY STATEMENT

The list of datasets and evaluation protocol are presented in Section 5.1. More comments and details on the dataset statistics are available in Appendix A. All hyperparameters can be found in Appendix C, full MRR and Hits@10 results with standard deviations are in Appendix D. The source code is available in the supplementary materials.

ACKNOWLEDGMENTS

This project is supported by Intel-Mila partnership program, the Natural Sciences and Engineering Research Council (NSERC) Discovery Grant, the Canada CIFAR AI Chair Program, collaboration grants between Microsoft Research and Mila, Samsung Electronics Co., Ltd., Amazon Faculty Research Award, Tencent AI Lab Rhino-Bird Gift Fund and a NRC Collaborative R&D Project (AI4D-CORE-06). This project was also partially funded by IVADO Fundamental Research Project grant PRF-2019-3583139727. The computation resource of this project is supported by Mila[4], Calcul Québec[5] and the Digital Research Alliance of Canada[6].

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

# A DATASETS

We conduct evaluation on 57 openly available KGs of various sizes and three groups, *i.e.*, tranducive, inductive with new entities, and inductive with both new entities and relations at inference time. The statistics for 16 transductive datasets are presented in Table 4, 18 inductive entity datasets in Table 5, and 23 inductive entity and relation datasets in Table 6. For each dataset, we also list a currently published state-of-the-art model that, at the moment, are all trained specifically on each target graph. Performance of those SOTA models is aggregated as *Supervised SOTA* in the results reported in the tables and figures. We omit smaller datasets (Kinships, UMLS, Countries, Family) with saturated performance as non-representative.

For the inductive datasets HM 1k, HM 3k, and HM 5k used in Hamaguchi et al. (2017) and Liu et al. (2021), we report the performance of predicting both heads and tails (noted as *b-1K*, *b-3K*, *b-5K* in Liu et al. (2021)) and compare against the respective baselines. Some inductive datasets (MT2, MT3, MT4) from MTDEA (Zhou et al., 2023) do not have reported entity-only KG completion performance. For Hetionet, we used the splits available in TorchDrug (Zhu et al., 2022b) and compare with the baseline RotatE reported by TorchDrug.

# B SPARSE MMS FOR RELATION GRAPH

The graph of relations $\mathcal{G}_r$ can be efficiently computed from the original multi-relational graph $\mathcal{G}$ with sparse matrix multiplications (spmm). Four spmm operations correspond to the four fundamental relation types $\{h2t, h2h, t2h, t2t\} \in \mathcal{R}_{fund}$.

Given the original graph $\mathcal{G}$ with $|\mathcal{V}|$ nodes and $|\mathcal{R}|$ relation types, its adjacency matrix is $\boldsymbol{A} \in \mathbb{R}^{|\mathcal{V}| \times |\mathcal{R}| \times |\mathcal{V}|}$. For clarity, $\boldsymbol{A}$ can be rewritten with *heads* $\mathcal{H}$ and *tails* $\mathcal{T}$ as $\boldsymbol{A} \in \mathbb{R}^{|\mathcal{H}| \times |\mathcal{R}| \times |\mathcal{T}|}$. From $\boldsymbol{A}$ we first build two sparse matrices $\boldsymbol{E}_h \in \mathbb{R}^{|\mathcal{H}| \times |\mathcal{R}|}$ and $\boldsymbol{E}_t \in \mathbb{R}^{|\mathcal{T}| \times |\mathcal{R}|}$ that capture the head-relation and tail-relation pairs, respectively. Computing interactions between relations is then equivalent to one spmm operation between relevant adjacencies:

$$\boldsymbol{A}_{h2h} = \text{spmm}(\boldsymbol{E}_h^T, \boldsymbol{E}_h) \in \mathbb{R}^{|\mathcal{R}| \times |\mathcal{R}|}$$
$$\boldsymbol{A}_{t2t} = \text{spmm}(\boldsymbol{E}_t^T, \boldsymbol{E}_t) \in \mathbb{R}^{|\mathcal{R}| \times |\mathcal{R}|}$$
$$\boldsymbol{A}_{h2t} = \text{spmm}(\boldsymbol{E}_h^T, \boldsymbol{E}_t) \in \mathbb{R}^{|\mathcal{R}| \times |\mathcal{R}|}$$
$$\boldsymbol{A}_{t2h} = \text{spmm}(\boldsymbol{E}_t^T, \boldsymbol{E}_h) \in \mathbb{R}^{|\mathcal{R}| \times |\mathcal{R}|}$$
$$\boldsymbol{A}_r = [\boldsymbol{A}_{h2h}, \boldsymbol{A}_{t2t}, \boldsymbol{A}_{h2t}, \boldsymbol{A}_{t2h}] \in \mathbb{R}^{|\mathcal{R}| \times |\mathcal{R}| \times 4}$$

For each of the four sparse matrices, the respective edge index is extracted from all non-zero values (or, similarly, by setting all non-zero values in the sparse matrix to ones). The final adjacency tensor of the graph of relations $\boldsymbol{A}_r$ and corresponding graph of relations $\mathcal{G}_r$ with four fundamental edge types can be obtained by stacking all four adjacencies ($[\cdot, \cdot]$ denotes stacking).

# C HYPERPARAMETERS

**Main results.** The hyperparameters for the pre-trained ULTRA model reported in Section 5.2 including Table 1, Table 2, Figure 1, and Figure 4 are presented in Table 7. Both GNNs over the relation graph $\mathcal{G}_r$ and main graph $\mathcal{G}$ are 6-layer GNNs with hidden dimension of 64, DistMult message function, and sum aggregation roughly following the NBFNet setup. Each layer of the $\text{GNN}_e$ (inductive link predictor over the main entity graph) features a 2-layer MLP as a function $g(\cdot)$ that transforms conditional relation representations into layer-specific relation representations. The model is trained on the mixture of FB15k237, WN18RR, and CoDEx-Medium graphs for 200,000 steps with batch size of 64 with AdamW optimizer and learning rate of 0.0005. Each batch contains only one graph and training samples from this graph. The sampling probability of the graph in the mixture is proportional to the number of edges in this training graph.

Table 4: Transductive datasets (16) used in the experiments. Train, Valid, Test denote triples in the respective set. Task denotes the prediction task: *h/t* is predicting both heads and tails, *tails* is only predicting tails. SOTA points to the best reported result.

| Dataset | Reference | Entities | Rels | Train | Valid | Test | Task | SOTA |
|---|---|---|---|---|---|---|---|---|
| CoDEx Small | Safavi & Koutra (2020) | 2034 | 42 | 32888 | 1827 | 1828 | h/t | ComplEx RP (Chen et al., 2021) |
| WDsinger | Lv et al. (2020) | 10282 | 135 | 16142 | 2163 | 2203 | h/t | LR-GCN (He et al., 2023) |
| FB15k237_10 | Lv et al. (2020) | 11512 | 237 | 27211 | 15624 | 18150 | tails | DacKGR (Lv et al., 2020) |
| FB15k237_20 | Lv et al. (2020) | 13166 | 237 | 54423 | 16963 | 19776 | tails | DacKGR (Lv et al., 2020) |
| FB15k237_50 | Lv et al. (2020) | 14149 | 237 | 136057 | 17449 | 20324 | tails | DacKGR (Lv et al., 2020) |
| FB15k237 | Toutanova & Chen (2015) | 14541 | 237 | 272115 | 17535 | 20466 | h/t | NBFNet (Zhu et al., 2021) |
| CoDEx Medium | Safavi & Koutra (2020) | 17050 | 51 | 185584 | 10310 | 10311 | h/t | ComplEx RP (Chen et al., 2021) |
| NELL23k | Lv et al. (2020) | 22925 | 200 | 25445 | 4961 | 4952 | h/t | LR-GCN (He et al., 2023) |
| WN18RR | Dettmers et al. (2018) | 40943 | 11 | 86835 | 3034 | 3134 | h/t | NBFNet (Zhu et al., 2021) |
| AristoV4 | Chen et al. (2021) | 44949 | 1605 | 242567 | 20000 | 20000 | h/t | ComplEx RP (Chen et al., 2021) |
| Hetionet | Himmelstein et al. (2017) | 45158 | 24 | 2025177 | 112510 | 112510 | h/t | RotatE (Sun et al., 2019) |
| NELL995 | Xiong et al. (2017) | 74536 | 200 | 149678 | 543 | 2818 | h/t | RED-GNN (Zhang & Yao, 2022) |
| CoDEx Large | Safavi & Koutra (2020) | 77951 | 69 | 551193 | 30622 | 30622 | h/t | ComplEx RP (Chen et al., 2021) |
| ConceptNet100k | Malaviya et al. (2020) | 78334 | 34 | 100000 | 1200 | 1200 | h/t | BiQUE (Guo & Kok, 2021) |
| DBpedia100k | Ding et al. (2018) | 99604 | 470 | 597572 | 50000 | 50000 | h/t | ComplEx-NNE+AER (Ding et al., 2018) |
| YAGO310 | Mahdisoltani et al. (2014) | 123182 | 37 | 1079040 | 5000 | 5000 | h/t | NBFNet (Zhu et al., 2021) |

Table 5: Inductive entity $(e)$ datasets (18) used in the experiments. Triples denote the number of edges of the graph given at training, validation, or test. Valid and Test denote triples to be predicted in the validation and test sets in the respective validation and test graph.

| Dataset | Rels | Training Graph | | Validation Graph | | | Test Graph | | | SOTA |
|---|---|---|---|---|---|---|---|---|---|---|
| | | Entities | Triples | Entities | Triples | Valid | Entities | Triples | Test | |
| FB v1 (Teru et al., 2020) | 180 | 1594 | 4245 | 1594 | 4245 | 489 | 1093 | 1993 | 411 | A*Net (Zhu et al., 2023) |
| FB v2 (Teru et al., 2020) | 200 | 2608 | 9739 | 2608 | 9739 | 1166 | 1660 | 4145 | 947 | NBFNet (Zhu et al., 2021) |
| FB v3 (Teru et al., 2020) | 215 | 3668 | 17986 | 3668 | 17986 | 2194 | 2501 | 7406 | 1731 | NBFNet (Zhu et al., 2021) |
| FB v4 (Teru et al., 2020) | 219 | 4707 | 27203 | 4707 | 27203 | 3352 | 3051 | 11714 | 2840 | A*Net (Zhu et al., 2023) |
| WN v1 (Teru et al., 2020) | 9 | 2746 | 5410 | 2746 | 5410 | 630 | 922 | 1618 | 373 | NBFNet (Zhu et al., 2021) |
| WN v2 (Teru et al., 2020) | 10 | 6954 | 15262 | 6954 | 15262 | 1838 | 2757 | 4011 | 852 | NBFNet (Zhu et al., 2021) |
| WN v3 (Teru et al., 2020) | 11 | 12078 | 25901 | 12078 | 25901 | 3097 | 5084 | 6327 | 1143 | NBFNet (Zhu et al., 2021) |
| WN v4 (Teru et al., 2020) | 9 | 3861 | 7940 | 3861 | 7940 | 934 | 7084 | 12334 | 2823 | A*Net (Zhu et al., 2023) |
| NELL v1 (Teru et al., 2020) | 14 | 3103 | 4687 | 3103 | 4687 | 414 | 225 | 833 | 201 | RED-GNN (Zhang & Yao, 2022) |
| NELL v2 (Teru et al., 2020) | 88 | 2564 | 8219 | 2564 | 8219 | 922 | 2086 | 4586 | 935 | RED-GNN (Zhang & Yao, 2022) |
| NELL v3 (Teru et al., 2020) | 142 | 4647 | 16393 | 4647 | 16393 | 1851 | 3566 | 8048 | 1620 | RED-GNN (Zhang & Yao, 2022) |
| NELL v4 (Teru et al., 2020) | 76 | 2092 | 7546 | 2092 | 7546 | 876 | 2795 | 7073 | 1447 | RED-GNN (Zhang & Yao, 2022) |
| ILPC Small (Galkin et al., 2022a) | 48 | 10230 | 78616 | 6653 | 20960 | 2908 | 6653 | 20960 | 2902 | NodePiece (Galkin et al., 2022a) |
| ILPC Large (Galkin et al., 2022a) | 65 | 46626 | 202446 | 29246 | 77044 | 10179 | 29246 | 77044 | 10184 | NodePiece (Galkin et al., 2022a) |
| HM 1k (Hamaguchi et al., 2017) | 11 | 36237 | 93364 | 36311 | 93364 | 1771 | 9899 | 18638 | 476 | R-GCN (Liu et al., 2021) |
| HM 3k (Hamaguchi et al., 2017) | 11 | 32118 | 71097 | 32250 | 71097 | 1201 | 19218 | 38285 | 1349 | Indigo (Liu et al., 2021) |
| HM 5k (Hamaguchi et al., 2017) | 11 | 28601 | 57601 | 28744 | 57601 | 900 | 23792 | 48425 | 2124 | Indigo (Liu et al., 2021) |
| IndigoBM (Liu et al., 2021) | 229 | 12721 | 121601 | 12797 | 121601 | 14121 | 14775 | 250195 | 14904 | GraIL (Liu et al., 2021) |

Table 6: Inductive entity and relation $(e, r)$ datasets (23) used in the experiments. Triples denote the number of edges of the graph given at training, validation, or test. Valid and Test denote triples to be predicted in the validation and test sets in the respective validation and test graph.

| Dataset | Training Graph | | | Validation Graph | | | | Test Graph | | | | SOTA |
|---|---|---|---|---|---|---|---|---|---|---|---|---|
| | Entities | Rels | Triples | Entities | Rels | Triples | Valid | Entities | Rels | Triples | Test | |
| FB-25 (Lee et al., 2023) | 5190 | 163 | 91571 | 4097 | 216 | 17147 | 5716 | 4097 | 216 | 17147 | 5716 | InGram (Lee et al., 2023) |
| FB-50 (Lee et al., 2023) | 5190 | 153 | 85375 | 4445 | 205 | 11636 | 3879 | 4445 | 205 | 11636 | 3879 | InGram (Lee et al., 2023) |
| FB-75 (Lee et al., 2023) | 4659 | 134 | 62809 | 2792 | 186 | 9316 | 3106 | 2792 | 186 | 9316 | 3106 | InGram (Lee et al., 2023) |
| FB-100 (Lee et al., 2023) | 4659 | 134 | 62809 | 2624 | 77 | 6987 | 2329 | 2624 | 77 | 6987 | 2329 | InGram (Lee et al., 2023) |
| WK-25 (Lee et al., 2023) | 12659 | 47 | 41873 | 3228 | 74 | 3391 | 1130 | 3228 | 74 | 3391 | 1131 | InGram (Lee et al., 2023) |
| WK-50 (Lee et al., 2023) | 12022 | 72 | 82481 | 9328 | 93 | 9672 | 3224 | 9328 | 93 | 9672 | 3225 | InGram (Lee et al., 2023) |
| WK-75 (Lee et al., 2023) | 6853 | 52 | 28741 | 2722 | 65 | 3430 | 1143 | 2722 | 65 | 3430 | 1144 | InGram (Lee et al., 2023) |
| WK-100 (Lee et al., 2023) | 9784 | 67 | 49875 | 12136 | 37 | 13487 | 4496 | 12136 | 37 | 13487 | 4496 | InGram (Lee et al., 2023) |
| NL-0 (Lee et al., 2023) | 1814 | 134 | 7796 | 2026 | 112 | 2287 | 763 | 2026 | 112 | 2287 | 763 | InGram (Lee et al., 2023) |
| NL-25 (Lee et al., 2023) | 4396 | 106 | 17578 | 2146 | 120 | 2230 | 743 | 2146 | 120 | 2230 | 744 | InGram (Lee et al., 2023) |
| NL-50 (Lee et al., 2023) | 4396 | 106 | 17578 | 2335 | 119 | 2576 | 859 | 2335 | 119 | 2576 | 859 | InGram (Lee et al., 2023) |
| NL-75 (Lee et al., 2023) | 2607 | 96 | 11058 | 1578 | 116 | 1818 | 606 | 1578 | 116 | 1818 | 607 | InGram (Lee et al., 2023) |
| NL-100 (Lee et al., 2023) | 1258 | 55 | 7832 | 1709 | 53 | 2378 | 793 | 1709 | 53 | 2378 | 793 | InGram (Lee et al., 2023) |
| Metafam (Zhou et al., 2023) | 1316 | 28 | 13821 | 1316 | 28 | 13821 | 590 | 656 | 28 | 7257 | 184 | NBFNet (Zhou et al., 2023) |
| FBNELL (Zhou et al., 2023) | 4636 | 100 | 10275 | 4636 | 100 | 10275 | 1055 | 4752 | 183 | 10685 | 597 | NBFNet (Zhou et al., 2023) |
| Wiki MT1 tax (Zhou et al., 2023) | 10000 | 10 | 17178 | 10000 | 10 | 17178 | 1908 | 10000 | 9 | 16526 | 1834 | NBFNet (Zhou et al., 2023) |
| Wiki MT1 health (Zhou et al., 2023) | 10000 | 7 | 14371 | 10000 | 7 | 14371 | 1596 | 10000 | 7 | 14110 | 1566 | NBFNet (Zhou et al., 2023) |
| Wiki MT2 org (Zhou et al., 2023) | 10000 | 10 | 23233 | 10000 | 10 | 23233 | 2581 | 10000 | 11 | 21976 | 2441 | N/A |
| Wiki MT2 sci (Zhou et al., 2023) | 10000 | 16 | 16471 | 10000 | 16 | 16471 | 1830 | 10000 | 16 | 14852 | 1650 | N/A |
| Wiki MT3 art (Zhou et al., 2023) | 10000 | 45 | 27262 | 10000 | 45 | 27262 | 3026 | 10000 | 45 | 28023 | 3113 | N/A |
| Wiki MT3 infra (Zhou et al., 2023) | 10000 | 24 | 21990 | 10000 | 24 | 21990 | 2443 | 10000 | 27 | 21646 | 2405 | N/A |
| Wiki MT4 sci (Zhou et al., 2023) | 10000 | 42 | 12576 | 10000 | 42 | 12576 | 1397 | 10000 | 42 | 12516 | 1388 | N/A |
| Wiki MT4 health (Zhou et al., 2023) | 10000 | 21 | 15539 | 10000 | 21 | 15539 | 1725 | 10000 | 20 | 15337 | 1703 | N/A |

**Fine-tuning and training from scratch.** Table 8 reports training durations for fine-tuning the pre-trained ULTRA and training models from scratch on each dataset (for the ablation study in Figure 5

Table 7: ULTRA hyperparameters for pre-training. GNN$_r$ denotes a GNN over the graph of relations $\mathcal{G}_r$, GNN$_e$ is a GNN over the original entity graph $\mathcal{G}$.

| | Hyperparameter | ULTRA pre-training |
|---|---|---|
| GNN$_r$ | # layers | 6 |
| | hidden dim | 64 |
| | message | DistMult |
| | aggeregation | sum |
| GNN$_e$ | # layers | 6 |
| | hidden dim | 64 |
| | message | DistMult |
| | aggregation | sum |
| | $g(\cdot)$ | 2-layer MLP |
| Learning | optimizer | AdamW |
| | learning rate | 0.0005 |
| | training steps | 200,000 |
| | adv temperature | 1 |
| | # negatives | 128 |
| | batch size | 64 |
| | Training graph mixture | FB15k237, WN18RR, CoDEx Medium |

and Section 5.3). In fine-tuning, if the number of fine-tuning epochs $k$ is more than one, we use the best checkpoint (out of $k$) evaluated on the validation set of the respective graph. Each fine-tuning run was repeated 5 times with different random seeds, each model trained from scratch was trained 3 times with different random seeds.

**Ablation: graphs in the training mixture.** For the ablation experiments reported in Figure 6, Table 9 describes the mixtures of graphs used in the pre-trained models. The mixtures of 5 and more graphs include large graphs of $100k+$ entities each, so we reduced the amount of training steps to complete training within 3 days (6 GPU-days in total as each model was trained on 2 A100 GPUs).

## D    FULL RESULTS

The full, per-dataset results of MRR and Hits@10 of the zero-shot inference of the pre-trained ULTRA model, the fine-tuned model, and best reported supervised SOTA baselines are presented in Table 10 and Table 11. The zero-shot results are deterministic whereas for fine-tuning performance we report the average of 5 different seeds with standard deviations.

Table 10 corresponds to Figure 1 and contains results on 43 graphs where published SOTA baselines are available, that is, on 3 pre-training graphs, on 14 inductive entity ($e$) graphs, on 13 inductive entity and relation ($e, r$) graphs, and 13 transductive graphs. Table 11 contains results on 16 graphs for which published SOTA exists only partially, that is, in terms of the Hits@10 (50 neg) metric computed against 50 randomly chosen negatives. We show that this metric greatly overestimates the real performance and encourage further works to report full MRR and Hits@k metrics computed against the whole entity set.

The results in Table 2 on 57 graphs (Section 5.2) are aggregated from Table 10 and Table 11.

Commenting on the performance of a pre-trained ULTRA model on larger transductive graphs, we attribute the performance difference to the following factors:

- Training data mixture and OOD generalization: the model reported in Table 1 was trained on 3 medium-sized KGs (15k - 40k nodes, 80k - 270k edges) while the biggest gaps are on larger graphs with many more nodes and edges (up to 120k nodes and 1M edges for YAGO 310), or many more relation types (1600+ in AristoV4), or very sparse (as in ConceptNet100k with 100k edges over 78k nodes). Size generalization issues are common for GNNs as found in Yehudai et al. (2021); Zhou et al. (2022). However, if we take the UL-

Table 8: Hyperparameters for fine-tuning ULTRA and training from scratch in the format (# epochs, steps per epoch), *e.g.*, (1, full) means one full epoch over the training set of the respective graph while (1, 1000) means 1 epoch of 1000 steps over the training set.

| Datasets | ULTRA fine-tuning | ULTRA train from scratch | Batch size |
|---|---|---|---|
| FB V1-V4 | (1, full) | (10, full) | 64 |
| WN V1-V4 | (1, full) | (10, full) | 64 |
| NELL V1-V4 | (3, full) | (10, full) | 64 |
| HM 1k-5k, IndigoBM | (1, 100) | (10, 1000) | 64 |
| ILPC Small | (3, full) | (10, full) | 64 |
| ILPC Large | (1, 1000) | (10, 1000) | 16 |
| FB 25-100 | (3, full) | (10, full) | 64 |
| WK 25-100 | (3, full) | (10, full) | 64 |
| NL 0-100 | (3, full) | (10, full) | 64 |
| MT1-MT4 | (3, full) | (10, full) | 64 |
| Metafam, FBNELL | (3, full) | (10, full) | 64 |
| WDsinger | (3, full) | (10, 1000) | 64 |
| NELL23k | (3, full) | (10, 1000) | 64 |
| FB237_10 | (1, full) | (10, 1000) | 64 |
| FB237_20 | (1, full) | (10, 1000) | 64 |
| FB237_50 | (1, 1000) | (10, 1000) | 64 |
| CoDEx-S | (1, 4000) | (10, 1000) | 64 |
| CoDEx-L | (1, 2000) | (10, 1000) | 16 |
| NELL-995 | (1, full) | (10, 1000) | 16 |
| YAGO 310 | (1, 2000) | (10, 2000) | 16 |
| DBpedia100k | (1, 1000) | (10, 1000) | 16 |
| AristoV4 | (1, 2000) | (10, 1000) | 16 |
| ConceptNet100k | (1, 2000) | (10, 1000) | 16 |
| Hetionet | (1, 4000) | (10, 1000) | 16 |
| WN18RR | (1, full) | (10, 1000) | 64 |
| FB15k237 | (1, full) | (10, 1000) | 64 |
| CoDEx-M | (1, 4000) | (10, 1000) | 64 |

Table 9: Graphs in different pre-training mixtures in Figure 6.

| | 1 | 2 | 3 | 4 | 5 | 6 | 8 |
|---|---|---|---|---|---|---|---|
| FB15k237 | ✓ | ✓ | ✓ | ✓ | ✓ | ✓ | ✓ |
| WN18RR | | ✓ | ✓ | ✓ | ✓ | ✓ | ✓ |
| CoDEx-M | | | ✓ | ✓ | ✓ | ✓ | ✓ |
| NELL995 | | | | ✓ | ✓ | ✓ | ✓ |
| YAGO 310 | | | | | ✓ | ✓ | ✓ |
| ConceptNet100k | | | | | | ✓ | ✓ |
| DBpedia100k | | | | | | | ✓ |
| AristoV4 | | | | | | | ✓ |
| Batch size | 32 | 16 | 64 | 16 | 16 | 16 | 16 |
| # steps | 200,000 | 400,000 | 200,000 | 400,000 | 200,000 | 200,000 | 200,000 |

TRA checkpoint pre-trained on 8 graphs (Table 9) and run evaluation on all 16 transductive graphs, then the average performance is better than supervised SOTA models, *i.e.*, 0.377 MRR / 0.537 Hits@10 of ULTRA against 0.371 MRR / 0.511 Hits@10 of the baselines.

• Transductive models have the privilege of memorizing target data distributions into entity/relation-specific vectors with overall many millions of parameters, *e.g.*, 80M parameters for a supervised SOTA BiQUE on ConceptNet100k. This performance, however, comes with the absence of transferability across KGs. In contrast, all pre-trained ULTRA checkpoints are rather small (about 170k parameters) but generalize to any KG. We ac-

knowledge the scaling behavior in the Section 6 and consider it a very promising avenue for future work. In particular, scaling laws for GNNs and common graph learning tasks (like link prediction) are not derived yet so we can only hypothesize whether there is any connection between GNNs size, dataset size, graph topology, and expected performance. Generally, there is no consensus in the graph learning community on whether deep or wide (non-geometric) GNNs bring immediate benefits - mostly due to the rising issues of over-smoothing and oversquashing (some initial results were recently presented in Di Giovanni et al. (2023)). In our experiments, we observe that the diversity of graphs in the pre-training mixture plays an important role as well. Therefore, we believe that a brute-force increase of the model size is unlikely to bring benefits unless paired with more diverse training data and more intricate mechanisms for capturing relational interactions.

## E    ON ADDING MORE FEATURES

Some graphs might have specific node and edge features such as numerical attributes and text descriptions. Often, KG features are heterogeneous, *e.g.*, graph from the life sciences domain would contain biomedical features that might not overlap with geographical features in other graphs, and would require different feature encoders. In the text domain, not all KGs have text features readily available as we mentioned in Section 2. In this work we focus on the structural representations and feature-less graphs as this can be applied to any KG with or without features.

Nevertheless, there is some evidence (Chen et al., 2022) that concatenating encoded text features (where available) to structural GNN features is likely to further boost the performance in inductive tasks. We consider dataset-specific features complementary to ULTRA representations and hypothesize that such additional features might be particularly useful at the fine-tuning stages. This is an intriguing direction for the future work.

## F    COMPUTATIONAL COMPLEXITY

The time complexity of ULTRA is upper-bounded by the entity-level $GNN_e$ (because the $GNN_r$ on the graph of relations has negligible overhead as the number of nodes in this graph is the same as number of unique relation types $|\mathcal{R}|$, and $|\mathcal{R}| \ll |\mathcal{V}|$, that is, the number of relation types is usually orders of magnitude smaller than the number of nodes). In our case, the main entity-level $GNN_e$ is NBFNet, so we mainly refer to the Appendix C of Zhu et al. (2021) for all necessary derivations.

The time complexity for a single layer is generally linear in the number of edges $O(|\mathcal{E}|d + |\mathcal{V}|d^2)$. With $T$ layers, the overall complexity of a single forward pass is $O(T(|\mathcal{E}|d + |\mathcal{V}|d^2))$ but $T$ is usually a small constant (6 layers) so the complexity is essentially linear to the number of edges. However, due to the sparsity of GNNs, they are usually bounded by memory. The memory complexity of the basic NBFNet implementation is $O(T|\mathcal{E}|d)$ and linear to the number of edges, but thanks to the efficient kernelized implementation of the relational message passing (already provided by NBFNet), the memory complexity is reduced to $O(T|\mathcal{V}|d)$ and is linear in the number of nodes. Moreover, the complexity can be further reduced when applying more scalable and optimized versions of entity-level GNNs such as AdaProp (Zhang et al., 2023) or A*Net (Zhu et al., 2023).

Table 10: Full results (MRR, Hits@10) of ULTRA in the zero-shot inference and fine-tuning regimes on 43 graphs compared to the best reported Supervised SOTA. The numbers correspond to Figure 1.

| Dataset | ULTRA 0-shot | | ULTRA fine-tuned | | Supervised SOTA | |
|---|---|---|---|---|---|---|
| | MRR | Hits@10 | MRR | Hits@10 | MRR | Hits@10 |
| *pre-training datasets* | | | | | | |
| WN18RR | | | 0.480 | 0.614 | 0.551 | 0.666 |
| FB15k237 | | | 0.368 | 0.564 | 0.415 | 0.599 |
| CoDEx Medium | | | 0.372 | 0.525 | 0.352 | 0.49 |
| *inductive ($e$) datasets* | | | | | | |
| WN V1 | 0.648 | 0.768 | 0.685 $\pm 0.003$ | 0.793 $\pm 0.003$ | 0.741 | 0.826 |
| WN V2 | 0.663 | 0.765 | 0.679 $\pm 0.002$ | 0.779 $\pm 0.003$ | 0.704 | 0.798 |
| WN V3 | 0.376 | 0.476 | 0.411 $\pm 0.008$ | 0.546 $\pm 0.006$ | 0.452 | 0.568 |
| WN V4 | 0.611 | 0.705 | 0.614 $\pm 0.003$ | 0.720 $\pm 0.001$ | 0.661 | 0.743 |
| FB V1 | 0.498 | 0.656 | 0.509 $\pm 0.002$ | 0.670 $\pm 0.004$ | 0.457 | 0.589 |
| FB V2 | 0.512 | 0.700 | 0.524 $\pm 0.003$ | 0.710 $\pm 0.004$ | 0.510 | 0.672 |
| FB V3 | 0.491 | 0.654 | 0.504 $\pm 0.001$ | 0.663 $\pm 0.003$ | 0.476 | 0.637 |
| FB V4 | 0.486 | 0.677 | 0.496 $\pm 0.001$ | 0.684 $\pm 0.001$ | 0.466 | 0.645 |
| NELL V1 | 0.785 | 0.913 | 0.757 $\pm 0.021$ | 0.878 $\pm 0.035$ | 0.637 | 0.866 |
| NELL V2 | 0.526 | 0.707 | 0.575 $\pm 0.004$ | 0.761 $\pm 0.007$ | 0.419 | 0.601 |
| NELL V3 | 0.515 | 0.702 | 0.563 $\pm 0.004$ | 0.755 $\pm 0.006$ | 0.436 | 0.594 |
| NELL V4 | 0.479 | 0.712 | 0.469 $\pm 0.020$ | 0.733 $\pm 0.011$ | 0.363 | 0.556 |
| ILPC Small | 0.302 | 0.443 | 0.303 $\pm 0.001$ | 0.453 $\pm 0.002$ | 0.130 | 0.251 |
| ILPC Large | 0.290 | 0.424 | 0.308 $\pm 0.002$ | 0.431 $\pm 0.001$ | 0.070 | 0.146 |
| *inductive ($e, r$) datasets* | | | | | | |
| FB-100 | 0.449 | 0.642 | 0.444 $\pm 0.003$ | 0.643 $\pm 0.004$ | 0.223 | 0.371 |
| FB-75 | 0.403 | 0.604 | 0.400 $\pm 0.003$ | 0.598 $\pm 0.004$ | 0.189 | 0.325 |
| FB-50 | 0.338 | 0.543 | 0.334 $\pm 0.002$ | 0.538 $\pm 0.004$ | 0.117 | 0.218 |
| FB-25 | 0.388 | 0.640 | 0.383 $\pm 0.001$ | 0.635 $\pm 0.002$ | 0.133 | 0.271 |
| WK-100 | 0.164 | 0.286 | 0.168 $\pm 0.005$ | 0.286 $\pm 0.003$ | 0.107 | 0.169 |
| WK-75 | 0.365 | 0.537 | 0.380 $\pm 0.001$ | 0.530 $\pm 0.009$ | 0.247 | 0.362 |
| WK-50 | 0.166 | 0.324 | 0.140 $\pm 0.010$ | 0.280 $\pm 0.012$ | 0.068 | 0.135 |
| WK-25 | 0.316 | 0.532 | 0.321 $\pm 0.003$ | 0.535 $\pm 0.007$ | 0.186 | 0.309 |
| NL-100 | 0.471 | 0.651 | 0.458 $\pm 0.012$ | 0.684 $\pm 0.011$ | 0.309 | 0.506 |
| NL-75 | 0.368 | 0.547 | 0.374 $\pm 0.007$ | 0.570 $\pm 0.005$ | 0.261 | 0.464 |
| NL-50 | 0.407 | 0.570 | 0.418 $\pm 0.005$ | 0.595 $\pm 0.005$ | 0.281 | 0.453 |
| NL-25 | 0.395 | 0.569 | 0.407 $\pm 0.009$ | 0.596 $\pm 0.012$ | 0.334 | 0.501 |
| NL-0 | 0.342 | 0.523 | 0.329 $\pm 0.010$ | 0.551 $\pm 0.012$ | 0.269 | 0.431 |
| *transductive datasets* | | | | | | |
| CoDEx Small | 0.472 | 0.667 | 0.490 $\pm 0.003$ | 0.686 $\pm 0.003$ | 0.473 | 0.663 |
| CoDEx Large | 0.338 | 0.469 | 0.343 $\pm 0.002$ | 0.478 $\pm 0.002$ | 0.345 | 0.473 |
| NELL-995 | 0.406 | 0.543 | 0.509 $\pm 0.013$ | 0.660 $\pm 0.006$ | 0.543 | 0.651 |
| YAGO 310 | 0.451 | 0.615 | 0.557 $\pm 0.009$ | 0.710 $\pm 0.003$ | 0.563 | 0.708 |
| WDsinger | 0.382 | 0.498 | 0.417 $\pm 0.002$ | 0.526 $\pm 0.002$ | 0.393 | 0.500 |
| NELL23k | 0.239 | 0.408 | 0.268 $\pm 0.001$ | 0.450 $\pm 0.001$ | 0.253 | 0.419 |
| FB15k237_10 | 0.248 | 0.398 | 0.254 $\pm 0.001$ | 0.411 $\pm 0.001$ | 0.219 | 0.337 |
| FB15k237_20 | 0.272 | 0.436 | 0.274 $\pm 0.001$ | 0.445 $\pm 0.002$ | 0.247 | 0.391 |
| FB15k237_50 | 0.324 | 0.526 | 0.325 $\pm 0.002$ | 0.528 $\pm 0.002$ | 0.293 | 0.458 |
| DBpedia100k | 0.398 | 0.576 | 0.436 $\pm 0.008$ | 0.603 $\pm 0.006$ | 0.306 | 0.418 |
| AristoV4 | 0.182 | 0.282 | 0.343 $\pm 0.006$ | 0.496 $\pm 0.004$ | 0.311 | 0.447 |
| ConceptNet100k | 0.082 | 0.162 | 0.310 $\pm 0.004$ | 0.529 $\pm 0.007$ | 0.320 | 0.553 |
| Hetionet | 0.257 | 0.379 | 0.399 $\pm 0.005$ | 0.538 $\pm 0.004$ | 0.257 | 0.403 |

Table 11: Full results (MRR, Hits@10) of ULTRA in the zero-shot inference and fine-tuning regimes on 14 graphs where Supervised SOTA reports an estimate Hits@10 (50 negs) metric (where available). The numbers correspond to Figure 4.

| Dataset | ULTRA 0-shot | | | ULTRA fine-tuned | | | Supervised SOTA |
|---|---|---|---|---|---|---|---|
| | MRR | Hits@10 | Hits@10 (50 neg) | MRR | Hits@10 | Hits@10 (50 neg) | Hits@10 (50 neg) |
| inductive $(e)$ datasets | | | | | | | |
| HM 1k | 0.059 | 0.092 | 0.796 | $0.042_{\pm 0.002}$ | $0.100_{\pm 0.007}$ | $0.839_{\pm 0.013}$ | 0.625 |
| HM 3k | 0.037 | 0.077 | 0.717 | $0.030_{\pm 0.002}$ | $0.090_{\pm 0.003}$ | $0.717_{\pm 0.016}$ | 0.375 |
| HM 5k | 0.034 | 0.071 | 0.694 | $0.025_{\pm 0.001}$ | $0.068_{\pm 0.003}$ | $0.657_{\pm 0.016}$ | 0.399 |
| IndigoBM | 0.440 | 0.648 | 0.995 | $0.432_{\pm 0.001}$ | $0.639_{\pm 0.002}$ | $0.995_{\pm 0.000}$ | 0.788 |
| inductive $(e, r)$ datasets | | | | | | | |
| MT1 tax | 0.224 | 0.305 | 0.731 | $0.330_{\pm 0.046}$ | $0.459_{\pm 0.056}$ | $0.994_{\pm 0.001}$ | 0.855 |
| MT1 health | 0.298 | 0.374 | 0.951 | $0.380_{\pm 0.002}$ | $0.467_{\pm 0.006}$ | $0.982_{\pm 0.002}$ | 0.858 |
| MT2 org | 0.095 | 0.159 | 0.778 | $0.104_{\pm 0.001}$ | $0.170_{\pm 0.001}$ | $0.855_{\pm 0.012}$ | - |
| MT2 sci | 0.258 | 0.354 | 0.787 | $0.311_{\pm 0.010}$ | $0.451_{\pm 0.042}$ | $0.982_{\pm 0.001}$ | - |
| MT3 art | 0.259 | 0.402 | 0.883 | $0.306_{\pm 0.003}$ | $0.473_{\pm 0.003}$ | $0.958_{\pm 0.001}$ | - |
| MT3 infra | 0.619 | 0.755 | 0.985 | $0.657_{\pm 0.008}$ | $0.807_{\pm 0.007}$ | $0.996_{\pm 0.000}$ | - |
| MT4 sci | 0.274 | 0.449 | 0.937 | $0.303_{\pm 0.007}$ | $0.478_{\pm 0.003}$ | $0.973_{\pm 0.001}$ | - |
| MT4 health | 0.624 | 0.737 | 0.955 | $0.704_{\pm 0.002}$ | $0.785_{\pm 0.002}$ | $0.974_{\pm 0.001}$ | - |
| Metafam | 0.238 | 0.644 | 1.0 | $0.997_{\pm 0.003}$ | $1.0_{\pm 0}$ | $1.0_{\pm 0}$ | 1.0 |
| FBNELL | 0.485 | 0.652 | 0.989 | $0.481_{\pm 0.004}$ | $0.661_{\pm 0.011}$ | $0.987_{\pm 0.001}$ | 0.95 |

