# OpenReview forum: "Towards Foundation Models for Knowledge Graph Reasoning"
_ICLR.cc/2024/Conference — ICLR 2024 poster_

### Official Review · Reviewer_sna7 · 2023-10-30

**Soundness:** 3 good
**Presentation:** 3 good
**Contribution:** 2 fair
**Rating:** 6
**Confidence:** 4

**Summary:**

This paper presents ULTRA, a single model that can be directly used/finetuned for link prediction over different knowledge graphs. The key is to model the transferrable relationships between different relations across knowledge graphs. Specifically, a NBFNet is used to learn relative relation representations and generate relation embeddings, which is then fed into another NBFNet to perform link predictions. Extensive experiments are performed over many knowledge graph to demonstrate the performance of this model.

**Strengths:**

- This paper is well written and easy to follow
- The core method around the relative relationships between relations is clever and interesting.
- The experiments demonstrate the gains of the method. It is especially impressive to see the competitive zero-shot performance of ULTRA over different knowledge graphs.

**Weaknesses:**

- The proposed method relies entirely on knowledge graph structure and does not consider using node embedding such as textual features of the knowledge graphs. In reality, text embedding of nodes and edges could be a better transferrable embedding. Such transferability has already been demonstrated by PRODIGY (https://arxiv.org/abs/2305.12600) and should be addressed.
- The model does not scale well as the authors already pointed out.
- The zero shot and fine-tuning performances are worse or on-par with the per dataset model performance, rendering pretraining not effective performance-wise.
- Some notations are a bit hard to understand. See questions.

**Questions:**

- What are u and v in h_{u|v} in section 4.2?
- Why are supervised SOTA baselines only reported for some datasets in Figure 4?

---

> ### Author Response · Authors · 2023-11-16
> **Response to sna7 Part 1**
>
> We thank the reviewer for highlighting the proposed method, impressive experimental results, and the quality of writing. Below, we would like to comment on the raised weaknesses and questions.
>
> > **W1. The proposed method relies entirely on knowledge graph structure and does not consider using node embedding such as textual features of the knowledge graphs. In reality, text embedding of nodes and edges could be a better transferrable embedding.**
>
> It is worth noting that LLM-processed text features are only available for KGs that do have those entity/relation descriptions available, but not all KGs have such text features readily available. In the case of numerical features, LLMs might be even a suboptimal encoder choice, eg, encoding numbers might require a different encoder, and if numerical features adhere to certain symmetries, then geometric models might be a better choice. That’s why in this work we focus on the structural representations and feature-less graphs as this can be applied to any KG with or without features. We briefly cover those reasons as well as some recent text-based approaches in the Related Work -> Text-based methods subsection.
>
> Having said that, there is some evidence (like in ReFactorGNNs [1]) that concatenating encoded text features (where available) to structural GNN features is likely to further boost the performance in inductive tasks. This is an intriguing direction for the future work and we elaborated on that in the new Appendix E in the updated version.
>
> Thank you for mentioning PRODIGY, it is certainly a related work (although focusing on different tasks of node classification and relation prediction) and we added it in the updated version of the manuscript.
>
> > **W2. The model does not scale well as the authors already pointed out.**
>
> We acknowledge the scaling behavior in the limitations and consider it a very promising avenue for future work. In particular, scaling laws for GNNs and common graph learning tasks (like link prediction) are not derived yet so we can only hypothesize whether there is any connection between GNNs size, dataset size, graph topology, and expected performance. Generally, there is no consensus in the graph learning community on whether deep or wide (non-geometric) GNNs bring immediate benefits - mostly due to the rising issues of oversmoothing and oversquashing (some initial results were recently presented in [2]). In our experiments, we observe that the diversity of graphs in the pre-training mixture plays an important role as well. Therefore, we believe that a brute-force increase of the model size is unlikely to bring benefits unless paired with more diverse training data and more intricate mechanisms for capturing relational interactions.
>
> > **W3. The zero shot and fine-tuning performances are worse or on-par with the per dataset model performance, rendering pretraining not effective performance-wise.**
>
> We would like to offer a different perspective on this matter (perhaps even the opposite) - if a single pre-trained model can deliver a comparable performance to 57 models trained individually from scratch on each dataset, then it saves a significant amount of compute needed to train those individual models. Zero-shot performance of ULTRA highlights such computational efficiency whereas fine-tuning requires only a fraction of the full training costs, as reported in Table 8 in the Appendix.
> Generally, we consider the ability of a single model to match the performance of numerous dataset-specific models to be promising. As we are at the beginning of the transfer learning era for KG reasoning, we believe that more fine-grained pre-training and fine-tuning techniques might further improve the compute/performance trade-off and it is an exciting area for the future work.
>
> > **Q1. What are $u$ and $v$ in $h_{u|v}$ in section 4.2?**
>
> Thanks for noticing that. Indeed, in Section 4.2 the notation of the indicator function and message passing can be simplified (we changed that in the updated version of the manuscript). In Section 4.2 which deals with the relations graph, the indicator function says that a node $v$ in the graph of relations will be initialized with some non-zero vector (vector of all ones in our case) if it’s equivalent to the query relation $q$ in the original input query $(h,q,?)$. The notation for a node $u$ is indeed redundant in this case as we can condition both equations on the query relation $q$ directly.

---

> > ### Author Response · Authors · 2023-11-16
> > **Response to sna7 Part 2 (final)**
> >
> > > **Q2. Why are supervised SOTA baselines only reported for some datasets in Figure 4?**
> >
> > The only available baseline for WikiTopics, Metafam, and FBNELL datasets is MTDEA [3] and INDIGO [4] for BM datasets. Both models have scalability issues and only report Hits@10 against 50 randomly chosen negatives. MTDEA was evaluated for the KG completion (entity prediction) task only on some of the graphs from the benchmark, ie, Metafam, FBNELL, WikiTopics MT1:tax and MT1:health. We compare with MTDEA and INDIGO in terms of this Hits@10 (50) metric where the baseline results are available.
> >
> > Since ULTRA is much more computationally efficient than both baselines, in addition to this limited metric, we also report ranking metrics (MRR and Hits@10) against the full entity set of each graph (which is a much harder task). The results indicate that evaluation against 50 random negatives overestimates the real performance and it is considerably lower when ranking against the full entity set.
> >
> > **References**
> > [1] Chen et al. ReFactor GNNs: Revisiting Factorisation-based Models from a Message-Passing Perspective. NeurIPS 2022.
> > [2] Di Giovanni et al. On Over-Squashing in Message Passing Neural Networks: The Impact of Width, Depth, and Topology. ICML 2023.
> > [3] Zhou et al. An OOD Multi-Task Perspective for Link Prediction with New Relation Types and Nodes. Arxiv 2023.
> > [4] Liu et al. INDIGO: GNN-Based Inductive Knowledge Graph Completion Using Pair-Wise Encoding. NeurIPS 2021.

---

> > > ### Comment · Reviewer_sna7 · 2023-11-22
> > >
> > > Thank you for the detailed response. I will maintain the current score.

---

### Official Review · Reviewer_HLbG · 2023-10-30

**Soundness:** 4 excellent
**Presentation:** 4 excellent
**Contribution:** 4 excellent
**Rating:** 8
**Confidence:** 3

**Summary:**

This work aims to build a foundation model for knowledge graph reasoning tasks, where the authors explore the setting of generalization to any edges and nodes, including unseen, of any multi-relational knowledge graphs without using node and edge features. To this end, the authors first construct a view of a relation-centric graph from an original graph where edges become nodes of this new relation graph, and then, based on this view, the authors represent the relation (node) relative to and conditioned on the query relation. Then, based on this relative relation representation, the authors use existing inductive link prediction methods to perform knowledge graph reasoning. The authors conduct link prediction experiments on various knowledge graphs considering both inductive and transductive settings, and show that the proposed method, namely ULTRA, outperforms other SOTA baselines sometimes without further fine-tuning on target knowledge graphs (i.e., zero-shot).

**Strengths:**

* This work studies the very important, challenging, and practical setups of building a foundation model for knowledge graph reasoning, which aims to be generalizable to any other knowledge graphs involving unseen nodes and unseen edges, without leveraging features of nodes and edges.
* The proposed method works well with different knowledge graphs, on zero-shot transfer learning setups without further fine-tuning on target knowledge graphs, and further shows the boosted performance with task-specific further fine-tuning on them, on most experiment setups.
* This paper is very well-written and easy to follow.

**Weaknesses:**

* I would like to note that I don't see any major weakness, and below is the minor.
* In Section 4.2, the explanation about the indicator function with variables $u$ and $v$ is a bit unclear to me. Could you elaborate more on the process and result of the indicator function according to those two variables, perhaps with visuals?
* Text-based methods (e.g., LM-based methods) can be generalizable to any knowledge graphs including unseen nodes and unseen edges, as long as their nodes and edges are represented with texts. In this vein, I think one potential direction for building a foundation model for knowledge graph-related tasks might be to use the LMs, and the authors may highlight this point more and potentially make comparisons between the proposed approach and text-based methods. I don't think this should be the critical weakness of this paper since text-based methods are limited to knowledge graphs with textual features; meanwhile, given the framing of this work ("Towards Foundation Models for Knowledge Graph Reasoning"), this point should be carefully explained.

**Questions:**

* I would like to suggest emphasizing the performance differences between inductive and transductive setups when explaining Table 1. The proposed method w/ 0-shot settings are strong on inductive graphs; meanwhile, previous methods are superior to it on transductive graphs, which are worthwhile to discuss.
* It may be beneficial to show the results of the ULTRA fine-tuned on the knowledge graphs used for pre-training the ULTRA. I am wondering if there are further performance improvements when further fine-tuning the model on the data used for pre-training.

---

> ### Author Response · Authors · 2023-11-16
> **Response to HLbG Part 1**
>
> We thank the reviewer for highlighting the merits of our work as to the importance of the problem, the ability to perform zero-shot inference and fine-tuning, as well as the quality of writing.
>
> Please find our comments on the weaknesses and questions below.
>
> > **W1. Could you elaborate more on the process and result of the indicator function according to those two variables, perhaps with visuals?**
>
> Indeed, in Section 4.2 the notation of the indicator function can be simplified (we changed that in the updated version of the manuscript). In Section 4.2 which deals with the graph of relations, the indicator function says that a node $v$ in the graph of relations will be initialized with some non-zero vector (vector of all ones in our case) if it’s equivalent to the query relation $q$ in the original input query $(h,q,?)$. The notation for a node $u$ is indeed redundant in this case as we can condition both equations on the query relation $q$ directly.
>
> > **W2. I think one potential direction … might be to use the LMs, and the authors may highlight this point more and potentially make comparisons between the proposed approach and text-based methods. I don't think this should be the critical weakness of this paper since text-based methods are limited to knowledge graphs with textual features;**
>
> You correctly pointed out that LLM-processed features are only available for KGs that do have those entity/relation features available, we briefly cover some recent approaches in the Related Work -> Text-based methods subsection. Not all KGs have such text features readily available and in the case of numerical features LLMs might be even a suboptimal encoder choice. That’s why in this work we focus on the structural representations and feature-less graphs as this can be applied to any KG with or without features.
>
> Having said that, there is some evidence (like in ReFactorGNNs [1]) that concatenating encoded text features (where available) to structural GNN features is likely to further boost the performance in inductive tasks. We consider dataset-specific features complementary to ULTRA representations and hypothesize that such additional features might be particularly useful at the fine-tuning stages. This is an intriguing direction for the future work and we added this discussion in the new Appendix E in the updated version.
>
> > **Q1. Previous methods are superior to ULTRA on transductive graphs, which are worthwhile to discuss.**
>
> Thank you for the suggestion, we added more discussion on the performance on larger transductive graphs in Appendix D. Generally, we attribute the performance difference to the following factors:
> * Training data mixture and OOD generalization: the model reported in Table 1 was trained on 3 medium-sized KGs (15k - 40k nodes, 80k - 270k edges) while the biggest gaps are on larger graphs with many more nodes and edges (up to 120k / 1M for YAGO 310), or many more relation types (1600+ in AristoV4), or very sparse (as in ConceptNet100k with 100k edges over 78k nodes). Size generalization issues are common for GNNs as found in [2] and [3]. However, if we take the ULTRA checkpoint pre-trained on 8 graphs (Table 9 in Appendix D) and run evaluation on all 16 transductive graphs, then the average performance is better than supervised SOTA models.
>
> |  | MRR | Hits@10 |
> | --- | --- | --- |
> | ULTRA (8 graphs) | 0.377 | 0.537 |
> | Supervised SOTA | 0.371 | 0.511 |
>
> Of course, for those datasets in the pretraining mix, we cannot call this regime a zero-shot inference anymore, but the results hint upon the hypothesis that more diverse graphs in the training mixture are beneficial for alleviating OOD issues.
> * Transductive models have the privilege of memorizing target data distributions (even on large graphs) into entity/relation-specific vectors with overall many millions of parameters, eg, 80M parameters for a supervised SOTA BiQUE on ConceptNet100k. This performance, however, comes with the absence of transferability across KGs. In contrast, all pre-trained ULTRA checkpoints are rather small (~170k parameters) but generalize to any KG. When it comes to scaling, scaling laws for GNNs and common graph learning tasks (like link prediction) are not derived yet so we can only hypothesize whether there is any connection between GNNs size, dataset size, graph topology, and expected performance. Generally, there is no consensus in the graph learning community on whether deep or wide (non-geometric) GNNs bring immediate benefits - mostly due to the rising issues of oversmoothing and oversquashing (some initial results were recently presented in [4]). In our experiments, we observe that the diversity of graphs in the pre-training mixture plays an important role as well. Therefore, we believe that a brute-force increase of the model size is unlikely to bring benefits unless paired with more diverse training data and more intricate mechanisms for capturing relational interactions.

---

> > ### Author Response · Authors · 2023-11-16
> > **Response to HLbG Part 2 (final)**
> >
> > > **Q2. I am wondering if there are further performance improvements when further fine-tuning the model on the data used for pre-training**
> >
> > We didn’t observe any significant performance improvements after fine-tuning the model specifically on the datasets used in the pre-training mixture of graphs.
> >
> > **References**:
> > [1] Chen et al. ReFactor GNNs: Revisiting Factorisation-based Models from a Message-Passing Perspective. NeurIPS 2022.
> > [2] Yehudai et al.  From local structures to size generalization in graph neural networks. ICML 2021.
> > [3] Zhou et al. OOD Link Prediction Generalization Capabilities of Message-Passing GNNs in Larger Test Graphs. NeurIPS 2022.
> > [4] Di Giovanni et al. On Over-Squashing in Message Passing Neural Networks: The Impact of Width, Depth, and Topology. ICML 2023.

---

> > > ### Comment · Reviewer_HLbG · 2023-11-18
> > >
> > > Thank you for responding to my comments and addressing all of them. After reading other reviews, I have no doubt that this is a good paper with significant contributions and it may be highlighted at the conference.

---

### Official Review · Reviewer_c3Sg · 2023-11-01

**Soundness:** 3 good
**Presentation:** 3 good
**Contribution:** 2 fair
**Rating:** 5
**Confidence:** 5

**Summary:**

Paper claims to propose a foundation model, named Ultra, for knowledge graph representation learning. The proposed model can handle full inductive graphs in which new entities and relations may appear in the test set. To do so, the authors propose to lift the graph to a one with relations as the nodes and design 4 different edge types (head2head, head2tail, tail2head, tail2tail). The relational representations are then learnt using message passing on this graph. The learnt relation embeddings are then used in the original graph to perform inductive link prediction. For the experiments, the authors pre-train their method on 3 KGs and further evaluate in a zero-shot setting and also by fine-tuning the downstream tasks.

**Strengths:**

- The paper proposes a transductive model that works in settings of new relations and entity nodes.
- The method obtains good zero-shot pretraining results.

**Weaknesses:**

- The authors have not explicitly stated the computational complexity of the method. From the paper, it seems that the forward pass is run on the entire relational graph to obtain relation representations. This is then used to initialize the node embedding from the query triple and the process is repeated for every triple. Thus it seems that the entire graph is being used for link prediction every triple making the computational complexity O(E^2). This seems limiting for large graphs that have not been explored in the paper (such as wikidata-5m etc.).
- From Table 2 we can see that finetuning over the pre-trained models helps the results significantly over the 0-shot setting. Also, the fine-tuning steps are too large to claim few shot results. This weakens the claim of the "foundation model" for KGs. A fair comparison would be to show the pretraining results for other inductive and transductive methods as well in addition to the SOTA comparison.
- Another limitation is that of scale. Since the current model has fewer parameters, this would limit learning over larger pretraining datasets as can be seen in Figure 6 and also reported by the authors.
- SoTA results for transductive models are better than the pre-trained Ultra model in many datasets. Thus the Ultra model seems to work well for the inductive setting rather than transductive. Thus the claim of the "foundation model" seems broader in scope.
- We see that in the metafam dataset, the pretraining results are poor but on finetuning the results are improved drastically. This shows that the method works well in cases where the relational patterns of the downstream datasets are similar to the pre-trained one but when the data distribution changes the results suffer. Moreover, due to limited capacity, the model may not be able to handle such cases by increasing the pretraining datasets calling for downstream finetuning. Thus domain adaptation is not a problem which can be easily overcome by scaling the current model and this further weakens the claim of a "foundation model" for KGs.

**Questions:**

- For weakness point 2: Any reason why this was not done by the authors?
- For weakness point 3: How would this be addressed in future works for the model?
-  For weakness point 4: Could the authors comment on why this would be the case and how would the model be improved to handle the transductive setting?
- For weakness point 5: Any reason why the results on this dataset are not good?
- Considering KGs are a rich source of textual/semantic data along with graph/structured data and the current model does not use this rich source of context information, how can we extend Ultra to incorporate the KG ontology?
- Considering weaknesses 2,4,5 the claim of the foundation model seems a bit broad as of now and at best the model could be said to be a good inductive learner.

---

> ### Author Response · Authors · 2023-11-16
> **Response to c3Sg Part 1**
>
> We thank the reviewer for the useful feedback and would like to comment on the weaknesses and questions.
>
> > **W1. The authors have not explicitly stated the computational complexity of the method.**
>
> Short answer: The time complexity is the same as of NBFNet, i.e., it is linear in the number of edges. The memory complexity is linear in the number of nodes when using a fused message passing kernel (already provided by NBFNet) or linear in the number of edges if materializing all edge messages without the kernel.
>
> Longer answer: The time complexity of ULTRA is upper-bounded by the entity-level GNN (because the GNN on the graph of relations has negligible overhead as the number of nodes in this graph is the same as number of unique relation types $|\mathcal{R}|$, and $|\mathcal{R}| \ll |\mathcal{V}|$ - the number of relation types is usually orders of magnitude smaller than the number of nodes, e.g, 37 relations in YAGO310 with 123k nodes). In our case, the main entity-level GNN is NBFNet, so we mainly refer to the Appendix C of the NBFNet paper [1] for all necessary derivations.
>
> The time complexity for a single layer is generally linear in the number of edges $O(|\mathcal{E}|d + |\mathcal{V}|d^2)$. With $T$ layers, the overall complexity of a single forward pass is $O(T(|\mathcal{E}|d + |\mathcal{V}|d^2))$ but $T$ is usually a small constant (6 layers) so the complexity is essentially linear to the number of edges.
> However, due to the sparsity of GNNs, they are usually bounded by memory. The memory complexity of the basic NBFNet implementation is $O(T|\mathcal{E}|d)$ and linear to the number of edges, but thanks to the efficient kernelized implementation of the relational message passing (already provided by NBFNet), the memory complexity is reduced to $O(T|\mathcal{V}|d)$ and is linear in the number of nodes.
>
> Moreover, the wall clock time and memory efficiency can be further reduced when applying more scalable and optimized versions of entity-level GNN such as AdaProp [2] or A*Net [3]. We include this discussion in the new Appendix F in the updated version of the manuscript.
>
> Predicting all possible triples in a graph is a rather artificial task and is hardly employed even by transductive shallow embedding methods as all such methods would require computing a $O(|\mathcal{V}| \times |\mathcal{R}| \times |\mathcal{V}|)$ tensor which is rather impractical.
>
> > **W2. A fair comparison would be to show the pretraining results for other inductive and transductive methods as well in addition to the SOTA comparison.**
>
> Short answer: we report the comparison with the only available pre-trainable baseline (InGram) in Table 3 dubbed as “no etypes, unconditional GNN (random)” and it is about 2x worse than ULTRA. The table caption includes the reference to InGram.
>
> Longer answer: The term “pre-training” cannot be applied to transductive models as they cannot run inductive inference on unseen graphs because their entity/relation embedding vocabularies are fixed to that particular graph. For example, a transductive RotatE trained on FB15k237 can only run inference on FB15k237 and not on other 50+ graphs in our benchmark.
> We are then left with inductive models which are then categorized into two families:
> * Inductive entity models, eg, NBFNet, RED-GNN, and similar) - they still learn relation embeddings pertaining to a particular data split and cannot be “pretrained” to run inference on unseen graphs, eg, NBFNet trained on FB V1 (one of the GraIL datasets from Teru et al) learns embeddings for 180 relations and can generalize to the test set of FB V1 where a graph has new entities but the same 180 relations. That said, such models still cannot generalize to unseen graphs with different relation vocabularies.
> * Inductive entity and relation models, eg, InGram, ISDEA, and MTDEA are the only models known in the literature at the moment. They can generalize to unseen relation vocabularies and can be “pretrained” to run inductive inference. Among them, ISDEA and MTDEA have issues with computational complexity and cannot scale to the graphs in our benchmark (eg, doing ranking evaluation on the full entity set). The only available baseline is InGram, and we do pretrain such a model on the same pre-training mixture as ULTRA and report its performance in Table 3 under the “no etypes, unconditional GNN (random)” ablation. InGram does not use edge types in the relation graph, does not use conditional GNN encoder, and uses random Glorot initialization for relations in the relation graph. Having the same pretraining mixture, such a model significantly underperforms ULTRA averaged across 54 graphs: 0.192 MRR vs 0.366 MRR of ULTRA.

---

> > ### Author Response · Authors · 2023-11-16
> > **Response to c3Sg Part 2**
> >
> > > **Also, the fine-tuning steps are too large to claim few shot results. This weakens the claim of the "foundation model" for KGs.**
> >
> > Could you please elaborate on this point? We do not have any few-shot claims in this paper and report a zero-shot performance and supervised fine-tuning performance (which is sample efficient and usually requires 2-4k gradient steps compared to full 10 epochs of training from scratch)
> >
> > > **W3. Another limitation is that of scale. Since the current model has fewer parameters, this would limit learning over larger pretraining datasets as can be seen in Figure 6 and also reported by the authors. How would this be addressed in future works for the model?**
> >
> > We acknowledge the scaling behavior in the limitations and consider it a very promising avenue for future work. In particular, scaling laws for GNNs and common graph learning tasks (like link prediction) are not derived yet so we can only hypothesize whether there is any connection between GNNs size, dataset size, graph topology, and expected performance. Generally, there is no consensus in the graph learning community on whether deep or wide (non-geometric) GNNs bring immediate benefits - mostly due to the rising issues of oversmoothing and oversquashing (some initial results were recently presented in [4]). In our experiments, we observe that the diversity of graphs in the pre-training mixture plays an important role as well. Therefore, we believe that a brute-force increase of the model size is unlikely to bring benefits unless paired with more diverse training data and more intricate mechanisms for capturing relational interactions.
> >
> > > **W4. SoTA results for transductive models are better than the pre-trained Ultra model in many datasets. Thus the Ultra model seems to work well for the inductive setting rather than transductive. Could the authors comment on why this would be the case and how would the model be improved to handle the transductive setting?**
> >
> > We added more discussion on the performance on larger transductive graphs in Appendix D. Generally, we attribute the performance difference to the following factors:
> > * Training data mixture and OOD generalization: the model reported in Table 1 was trained on 3 medium-sized KGs (15k - 40k nodes, 80k - 270k edges) while the biggest gaps are on larger graphs with many more nodes and edges (up to 120k / 1M for YAGO 310), or many more relation types (1600+ in AristoV4), or very sparse (as in ConceptNet100k with 100k edges over 78k nodes). Size generalization issues are common for GNNs as found in [5] and [6]. However, if we take the ULTRA checkpoint pre-trained on 8 graphs (Table 9 in Appendix D) and run evaluation on all 16 transductive graphs, then the average performance is better than supervised SOTA models.
> >
> > |  | MRR | Hits@10 |
> > | --- | --- | --- |
> > | ULTRA (8 graphs) | 0.377 | 0.537 |
> > | Supervised SOTA | 0.371 | 0.511 |
> >
> > Of course, for those 8 graphs in the training mixture, we cannot call this regime a zero-shot inference anymore, but the results hint upon the hypothesis that more diverse graphs in the training mixture are beneficial for alleviating OOD issues.
> >
> > * Transductive models have the privilege of memorizing target data distributions (even on large graphs) into entity/relation-specific vectors with overall many millions of parameters, eg, 80M parameters for BiQUE on ConceptNet100k. This performance, however, comes with the absence of transferability across KGs. In contrast, all pre-trained ULTRA checkpoints are rather small (~170k parameters) but generalize to any KG. As we discussed in the previous paragraphs and answer to W3, the question of scaling inductive GNNs is still open and there might be many contributing factors to it.
> >
> > > **W5. Metafam: the pretraining results are poor, improved after fine-tuning. The method works ... when relational patterns of the downstream datasets are similar to the pre-trained ones but otherwise suffer. why the results on this dataset are not good?**
> >
> > Metafam proposed in the MTDEA paper [7] is a synthetic dataset that does not have a single connected graph in train/test splits. To be precise, the dataset has 50 disconnected components in the training split and 25 disconnected components in the test split, in both splits the size of each connected component is capped by 26 nodes. This is the only synthetic dataset among 57 graphs considered in this work whereas the other graphs are derived from more natural graphs such as Wikidata, Freebase, or DBpedia. With those reasons in mind, we would rather refrain from drawing generalization conclusions based on the single synthetic dataset in light of the results on the other 50+ graphs. As we are at the beginning of exploring transferable invariances on KGs, it is expected that no model is optimal for all tasks in the zero-shot inference mode. Still, a short fine-tuning allows ULTRA to quickly adapt to this setup with many disconnected components and solve the task almost perfectly.

---

> > > ### Author Response · Authors · 2023-11-16
> > > **Response to c3Sg Part 3 (final)**
> > >
> > > > **Q6. Considering KGs are a rich source of textual/semantic data along with graph/structured data and the current model does not use this rich source of context information, how can we extend Ultra to incorporate the KG ontology?**
> > >
> > > KG features are often heterogeneous, e.g., life sciences graphs would contain biomedical features that do not overlap with geographical features in other graphs, and would require different feature encoders. In the text domain, not all KGs have text features readily available. That’s why in this work we focus on the structural representations and feature-less graphs as this can be applied to any KG with or without features. We briefly cover those reasons as well as some recent text-based approaches in the Related Work -> Text-based methods subsection.
> > >
> > > Having said that, there is some evidence (like in ReFactorGNNs [8]) that concatenating encoded text features (where available) to structural GNN features is likely to further boost the performance in inductive tasks. This is an intriguing direction for the future work and we elaborated on that in the new Appendix E in the updated version.
> > >
> > > Finally, ontologies are limited to a particular relation vocabulary (that is, there is no unified ontology combining all 50+ graphs in our benchmark) and provide only deductive knowledge which is rather orthogonal to the inductive reasoning task we focus on in this work. We hypothesize that ontology information might be useful at the supervised fine-tuning stage and would envision that as one of the areas for the future work.
> > >
> > > **References:**
> > > [1] Zhu et al. Neural Bellman-Ford Networks: A General Graph Neural Network Framework for Link Prediction. NeurIPS 2021
> > > [2] Zhang et al. AdaProp: Learning Adaptive Propagation for Graph Neural Network based Knowledge Graph Reasoning. KDD 2023
> > > [3] Zhu et al. A*Net: A Scalable Path-based Reasoning Approach for Knowledge Graphs. NeurIPS 2023
> > > [4] Di Giovanni et al. On Over-Squashing in Message Passing Neural Networks: The Impact of Width, Depth, and Topology. ICML 2023
> > > [5] Yehudai et al.  From local structures to size generalization in graph neural networks. ICML 2021
> > > [6] Zhou et al. OOD Link Prediction Generalization Capabilities of Message-Passing GNNs in Larger Test Graphs. NeurIPS 2022
> > > [7] Zhou et al. An OOD Multi-Task Perspective for Link Prediction with New Relation Types and Nodes. Arxiv 2023
> > > [8] Chen et al. ReFactor GNNs: Revisiting Factorisation-based Models from a Message-Passing Perspective. NeurIPS 2022

---

> ### Comment · Reviewer_c3Sg · 2023-11-22
>
> Thank you for the detailed reply and also for the additional experiments.
>
> The responses address some of my concerns.
>
> Before updating my score, I have further clarification questions/suggestions as below:
>
> 1) For the baselines /related works for inductive relation/link prediction there are a few additional works [1,2,3,4,5] that the authors may have missed. It would help to discuss these in the paper.
>
> 2) One concern that remains is that the method still needs to be finetuned on the entire dataset in most cases. I get that ULTRA requires 1 epoch (or thousands of steps) vs 10 epochs from scratch. But would it not be expected from a foundation model that the finetuning steps required should be orders of magnitude lesser? It would also help to have this study of the performance of ULTRA with a few examples (in addition to zero-shot performance) to understand the nature of the model in a new setting. This may also help practitioners decide how much training is required for the new dataset.
>
> References:
>
> 1] https://proceedings.neurips.cc/paper_files/paper/2021/hash/a1c5aff9679455a233086e26b72b9a06-Abstract.html
>
> 2] https://proceedings.mlr.press/v162/yan22a.html
>
> 3] https://ieeexplore.ieee.org/abstract/document/9534355
>
> 4] https://link.springer.com/article/10.1007/s11280-023-01168-w
>
> 5] https://ojs.aaai.org/index.php/AAAI/article/view/16779

---

> > ### Author Response · Authors · 2023-11-23
> > **Response**
> >
> > 1. Thank you for the reference suggestions, we will include those into the final version in the relevant subsections. It’s worth mentioning that [1] is not quite about link prediction, while [2][3][4][5] are tackling the inductive entity setup only, ie, not transferring between unseen relations.
> >
> > 2. There is always a trade-off between fine-tuning time and performance, and it's possible to sacrifice one in favor of the other. In our work, fine-tuning is optional (as zero-shot inference results on average are already better than supervised baselines), so we are interested in maximizing the gains after fine-tuning. Generally, fine-tuning practices of BERT made it rather common to fine-tune for 2-5 epochs (or a few thousand steps), and we aimed at about the same range (adjusted by sample efficiency of labeling trick GNNs, dataset sizes, and the non-LLM nature of transferable invariances considered in our work). The main ULTRA model with results reported in the figures was pre-trained for 200.000 steps (please refer to Table 9 for the info on the other pre-trained models). Graph-specific fine-tuning takes 2000-4000 steps which is indeed orders of magnitude less – exactly as you mentioned (up to 100x less). Please note that on smaller datasets with <50k triples, one fine-tuning epoch corresponds to the same 1500 (batch size 32) – 4000 (batch size 16) steps depending on the batch size.
> >
> > In contrast to the main models pre-trained on several (2-8) graphs, ablated models trained from scratch on a single dataset saturate rather quickly, so there is no significant performance boost in training them any longer.
> >
> > > This may also help practitioners decide how much training is required for the new dataset.
> >
> > As a rule of thumb, we would suggest 1 epoch for graphs with less than 50k triples (with batch size 32 this is equivalent to about 1500 steps) and 2000 steps for anything larger.

---

### Official Review · Reviewer_ebFz · 2023-11-02

**Soundness:** 3 good
**Presentation:** 3 good
**Contribution:** 2 fair
**Rating:** 8
**Confidence:** 4

**Summary:**

The key limitation of designing the foundation models for dealing with the Knowledge Graphs (KGs) is that the KGs have different entities and relations that generally do not overlap. To address this issue, this paper proposes ULTRA, which positively transfers the information of source KG to unseen KG. It constructs relation representations based on the interactions between the relations by introducing the graph of relations. The proposed approach has shown good performance on various tasks.

**Strengths:**

- From their experiments, the proposed methods have shown good performance on various tasks.
- Research topics about the foundational models on graph-structured datasets is really interesting and important.
- The paper is well written and easy to follow.

**Weaknesses:**

- The authors first pretrain the ULTRA model with the mixture of 3 standard KGs and then fine the model for the downstream task. But, the other supervised SOTA model only uses dataset of the downstream tasks without employing the pre-training datasets. If the supervised SOTA models are designed to deal with transductive settings, they may show worse performance on the downstream tasks. However, if the SOTA models are the models for the inductive setting, I think they may be possible to be pretrained like the ULTRA. So, could you measure the performance of the "pretrained" SOTA models on the inductive if possible?

**Questions:**

Please refer to the weaknesses.

---

> ### Author Response · Authors · 2023-11-16
> **Response to ebFz**
>
> We thank the reviewer for acknowledging the merits of our work including the importance of the tackled problem, experimental results, and writing quality.
> Below, we would like to comment on the identified weaknesses.
>
> > **If the supervised SOTA models are designed to deal with transductive settings, they may show worse performance on the downstream tasks**
>
> Yes, this is correct. To be even more precise, transductive models trained on one KG cannot run inductive inference on downstream tasks at all because their entity/relation embedding vocabularies are fixed to that particular graph. For example, a transductive RotatE trained on FB15k237 can only run inference on FB15k237 and not on other 50+ graphs in our benchmark. In that sense, the term “pre-training” cannot be applied to transductive models as they only fit one particular graph data distribution with a fixed set of entities and relations.
>
> > **So, could you measure the performance of the "pretrained" SOTA models on the inductive if possible?**
>
> Short answer: we report the comparison with the only available baseline (InGram) in Table 3 dubbed as “no etypes, unconditional GNN (random)” and it is about 2x worse than ULTRA. The table caption includes the reference to InGram.
>
> Elaborated answer:
> Based on the first paragraph, we cannot pretrain transductive SOTA models to run inductive inference on our benchmark. We are then left with inductive models which are then categorized into two families:
> * Inductive entity models, eg, NBFNet, RED-GNN, and similar) - they still learn relation embeddings pertaining to a particular data split and cannot be “pretrained” to run inference on unseen graphs, eg, NBFNet trained on FB V1 (one of the GraIL datasets from Teru et al) learns embeddings for 180 relations and can generalize to the test set of FB V1 where a graph has new entities but the same 180 relations. That said, such models still cannot generalize to unseen graphs with different relation vocabularies.
> * Inductive entity and relation models, eg, InGram, ISDEA, and MTDEA are the only models known in the literature at the moment. They can generalize to unseen relation vocabularies and can be “pretrained” to run inductive inference. Among them, ISDEA and MTDEA have issues with computational complexity and cannot scale to the graphs in our benchmark (eg, doing ranking evaluation on the full entity set). The only available baseline is InGram, and we do pretrain such a model on the same pre-training mixture as ULTRA and report its performance in Table 3 under the “no etypes, unconditional GNN (random)” ablation. InGram does not use edge types in the relation graph, does not use conditional GNN encoder, and uses random Glorot initialization for relations in the relation graph. Having the same pretraining mixture, such a model significantly underperforms ULTRA averaged across 54 graphs: 0.192 MRR vs 0.366 MRR of ULTRA.

---

> > ### Comment · Reviewer_ebFz · 2023-11-22
> >
> > Thank you for the response. I have read all the reviews and their corresponding responses.
> >
> > My all concerns are addressed and I keep my rating.

---

### Author Response · Authors · 2023-11-16
**General Response**

We thank the reviewers for the feedback and useful comments. In addition to detailed responses, in this general comment, we would like to provide a summary of identified strengths and changes in the new revision addressing common weaknesses. The new additions in the manuscript are marked in blue.

Strengths:
* **Task importance and method novelty:** “*The work studies the very important, challenging, and practical setups of building a foundation model for knowledge graph reasoning … without features of nodes and edges*” (Reviewer HLbG). “*Research … about the foundational models on graph-structured datasets is really interesting and important*” (Reviewer ebFz). “*The core method is clever and interesting*” (Reviewer sna7).
* **Experimental results** (highlighted by all reviewers): “*It is especially impressive to see the competitive zero-shot performance of ULTRA over different knowledge graphs.*” (Reviewer sna7) “*The proposed method works well with different knowledge graphs, on zero-shot setups and … with task-specific further fine-tuning on them*” (HLbG). “*The method obtains good zero-shot pretraining results.*” (Reviewer c3Sg). “*The proposed methods have shown good performance on various tasks.*” (Reviewer ebFz)
* **Writing quality, the paper is easy to follow** (Reviewers ebFz, HLbG, sna7)

In the updated manuscript, we:
* Added **Appendix F** discussing the time and memory complexity of our approach that is similar to NBFNet (Reviewer c3Sg)
* Added **Appendix E** on the **complementary nature of structural ULTRA representations and additional textual/numerical features**(Reviewers c3Sg, HLbG, sna7)
* Expanded **Appendix D** to discuss (1) the performance gap on larger transductive graphs and its absence when adding those graphs to the training mixture (Reviewers c3Sg, HLbG); and (2) paths to better parameter scaling of the model (Reviewers c3Sg, sna7)
* Simplified the notation in Section 4.2 (Reviewers HLbG, sna7)
* Improved the Related Work section

---

### Meta-Review · Area_Chair_pcsJ · 2023-12-06

**Metareview:**

This paper presents a foundation model, named Ultra, for knowledge graph reasoning tasks. The paper studies a timely and important problem, and results are quite promising. The paper is well written and clearly organized. Reviewers raised some questions regarding technical details, results, scalability, complexity, etc. which have been mostly addressed in the rebuttal. The authors are highly encouraged to incorporate the suggestions from reviewers into the final version of the paper.

**Justification For Why Not Higher Score:**

The paper has some limitations such as scalability.

**Justification For Why Not Lower Score:**

The paper studies an important problem and presents a convincing solution.

---

### Decision · Program_Chairs · 2024-01-16

Accept (poster)